# Network connectivity determines cortical thinning in early Parkinson's disease progression

Y. Yau[1], Y. Zeighami[1], T.E. Baker[1,2], K. Larcher[1], U. Vainik [1,3], M. Dadar[1], V.S. Fonov[1], P. Hagmann[4], A. Griffa[5], B. Mišić[1], D.L. Collins[1] & A. Dagher [1]

Here we test the hypothesis that the neurodegenerative process in Parkinson's disease (PD) moves stereotypically along neural networks, possibly reflecting the spread of toxic alpha-synuclein molecules. PD patients ($n = 105$) and matched controls ($n = 57$) underwent T1-MRI at entry and 1 year later as part of the Parkinson's Progression Markers Initiative. Over this period, PD patients demonstrate significantly greater cortical thinning than controls in parts of the left occipital and bilateral frontal lobes and right somatomotor-sensory cortex. Cortical thinning is correlated to connectivity (measured functionally or structurally) to a "disease reservoir" evaluated by MRI at baseline. The atrophy pattern in the ventral frontal lobes resembles one described in certain cases of Alzheimer's disease. Our findings suggest that disease propagation to the cortex in PD follows neuronal connectivity and that disease spread to the cortex may herald the onset of cognitive impairment.

[1] Montreal Neurological Institute, McGill University, 3801 University Street, Montreal, QC H3A 2B4, Canada. [2] Center for Molecular and Behavioral Neuroscience, Rutgers University, 197 University Avenue, Newark, NJ 07102, USA. [3] Institute of Psychology, Faculty of Social Sciences, University of Tartu, Näituse 2, 50409 Tartu, Estonia. [4] Department of Radiology, Lausanne University Hospital and University of Lausanne, Rue du Bugnon 21, 1011 Lausanne, Switzerland. [5] Brain Center Rudolf Magnus UMC Utrecht Heidelberglaan 100, A01.126 3508 GA Utrecht, The Netherlands. Y. Yau and Y. Zeighami contributed equally to this work. Correspondence and requests for materials should be addressed to A.D. (email: alain.dagher@mcgill.ca)

The network-spread hypothesis proposes that neurodegenerative diseases target intrinsic brain networks. Specifically, it is suggested that the progression of disease follows network connectivity and that neurotoxicity is caused by the spread and accumulation of toxic agents along the brain's neuronal connectome[1,2]. In the case of Parkinson's disease (PD), the likely agent is misfolded α-synuclein (α-syn). In its normal conformation, α-syn is an endogenous protein that is soluble and contributes to synaptic vesicular function[3]; however, in synucleinopathies such as PD, α-syn is present in a hyperphosphorylated and insoluble state and found to adopt an abnormal configuration into amyloid fibril aggregates[4]. A model first proposed by Braak suggested that patterns of abnormal α-syn accumulation observed at postmortem were consistent with a stereotyped propagation from brainstem to subcortical areas to cortex, which paralleled the clinical progression of the disease[5]. Numerous lines of evidence support this model: misfolded α-syn aggregates are identified at postmortem in the brains of PD patients, and the protein has been shown to spread through neuroanatomical pathways causing cell death in animal models due to its neurotoxicity[4]; however, the model remains controversial and direct human evidence is sparse[6,7].

There is currently no imaging technique that allows in vivo measurement of α-syn in the brain. However, we previously observed a pattern of brain atrophy in cross-sectional magnetic resonance imaging (MRI) scans of de novo PD patients from the Parkinson's Progression Markers Initiative (PPMI) database[8] that was consistent with the network-spread hypothesis[9]. The regional distribution of atrophy identified using deformation-based morphometry (DBM) overlapped with an intrinsic connectivity network, and the pattern suggested a disease epicenter in the substantia nigra. The atrophy distribution was mostly subcortical, involving brainstem, striatum, thalamus, and medial temporal lobe.

Recent studies using voxel-based morphometry (VBM) to assess gray matter density have begun to highlight a pattern of cortical changes in PD[10,11], but there have been discrepant findings[12,13]. Inconsistencies are not only due to heterogeneity in disease progression and small sample sizes, but also to methodological issues[14]. Cortical thickness analysis may be better able to delineate cortical morphometrical patterns and may be more sensitive to age or disease-related regional gray matter changes than VBM[15,16]. While cortical thinning is normal in the aging process[17], deviations from typical patterns can be used as diagnostic indicators for neurodegenerative disorders (e.g., Alzheimer's disease (AD)) even at early stages[18]. Several cross-sectional studies of MRI-derived cortical thickness in early-stage PD have reported cortical thinning in parietal and premotor regions compared to controls[16,19–21]. Moreover, cortical thinning appears to correlate with poorer cognitive performance[22]. However, cortical thickness in PD has yet to be studied in a controlled longitudinal setting, and little is known as to how it may correlate with other disease-related measures.

Disease spread to cortical areas may be accompanied by the start of cognitive impairment. Dementia is now recognized as a common, perhaps inevitable, feature of PD[23,24]. Cognitive impairment takes several forms, starting with executive (frontal lobe) dysfunction and apathy, and progressing to visual hallucinations, impulse control, and behavioral disorders, and, eventually, a global cognitive impairment that resembles AD[24–27]. As with the classical dementias, it is likely that the specific pattern of disease spread in PD may shape the clinical features of the dementing process[1].

Here we take advantage of follow-up imaging in a large longitudinal, multicenter study that aimed to assess the progression of clinical features, imaging, and other biomarkers in a cohort of well-characterized de novo PD patients. Using the PPMI database (http://www.ppmi-info.org/data), we test the theory that disease progression at a 1-year follow-up, as assessed by cortical thickness, is best explained by spread from a "disease reservoir" along the brain's connectome. We further identify certain factors at baseline that predict severity of disease progression at follow-up and provide evidence that cortical involvement might herald the onset of cognitive decline.

## Results

**Participants.** Participants in the present study consisted of de novo PD patients ($n = 105$) and healthy controls (HC, $n = 57$) enrolled in the PPMI study[8] [accessed May 01, 2016], clinicaltrials.gov identifier NCT01141023] who had follow-up assessment 1-year postenrollment (1.05 years for both groups). Demographic and clinical data are summarized in Table 1. PD patients were unmedicated at the first evaluation. At 1-year follow-up, 81 of the 105 were on PD medication (e.g., levodopa or dopamine agonist). Among PD patients, performance on the Montreal Cognitive Assessment (MoCA) was slightly worse at follow-up compared to baseline (from 27.5 to 26.7, $t(103) = 3.038$, $p = 0.003$). PD patients also scored higher on the Movement Disorders Society Unified Parkinson's Disease Rating Scale Part III (UPDRS-III) at follow-up compared to baseline, but this did not reach statistical significance (from 21.7 to 23.1, $t(95) = -1.544$, $p = 0.126$).

**Changes in cortical thickness at 1-year follow-up.** Over the 1-year period, mean whole-brain cortical thickness significantly decreased among both PD patients ($t_1 = 3.055$ mm $\pm 0.013$; $t_2 = 3.027$ mm $\pm 0.013$) and controls ($t_1 = 3.055$ mm $\pm 0.016$; $t_2 = 3.036$ mm $\pm 0.017$) (both $p < 0.01$). Regional cortical thinning was greater in PD than HC (Fig. 1). Specifically, four significant clusters were identified: (1) left occipital and inferior and middle temporal gyri, (2) left frontal, (3) right frontal, and (4) right pre-

### Table 1 Demographic, clinical, and laboratory measures

| | PD patients | Healthy controls | t-test/$\chi^2$ | p-value |
|---|---|---|---|---|
| Age | | | | |
| $t_1$ | 61.06 ± 9.37 | 59.05 ± 10.96 | 1.224 | 0.223 |
| $t_2$ | 62.11 ± 9.37 | 60.10 ± 10.96 | 1.223 | 0.223 |
| Sex | 33 F: 72 M | 22 F: 35 M | 0.846 | 0.358 |
| UPDRS-III | | | | |
| $t_1$ | 21.67 ± 9.61 | — | — | — |
| $t_2$ | 23.10 ± 10.67 | — | — | — |
| MoCA | | | | |
| $t_1$ | 27.49 ± 2.11 | 28.30 ± 1.13 | −3.162 | 0.002 |
| $t_2$ | 26.67 ± 2.78 | 27.35 ± 2.01 | −1.778 | 0.077 |
| α-syn | | | | |
| $t_1$ | 1841.54 ± 729.73 | 1898.84 ± 753.12 | −0.462 | 0.645 |
| Aβ42 | | | | |
| $t_1$ | 370.33 ± 97.02 | 373.36 ± 92.53 | −0.189 | 0.850 |
| t-tau | | | | |
| $t_1$ | 44.77 ± 18.36 | 46.06 ± 19.69 | −0.407 | 0.684 |
| p-tau$_{181}$ | | | | |
| $t_1$ | 15.85 ± 9.74 | 17.30 ± 9.84 | −0.885 | 0.377 |

$t_1$ measures at entry, $t_2$ measures at 1-year follow-up. Values are expressed as mean ± standard deviation. Two rightmost columns refer to the difference between PD and controls. All cerebrospinal fluid measures are expressed in pg/ml
UPDRS-III Movement Disorders Society Unified Parkinson's Disease Rating Scale Part III, MoCA Montreal Cognitive Assessment, α-syn α-synuclein, Aβ42 β-amyloid 1–42, t-tau total tau, p-tau$_{181}$ phosphorylated tau

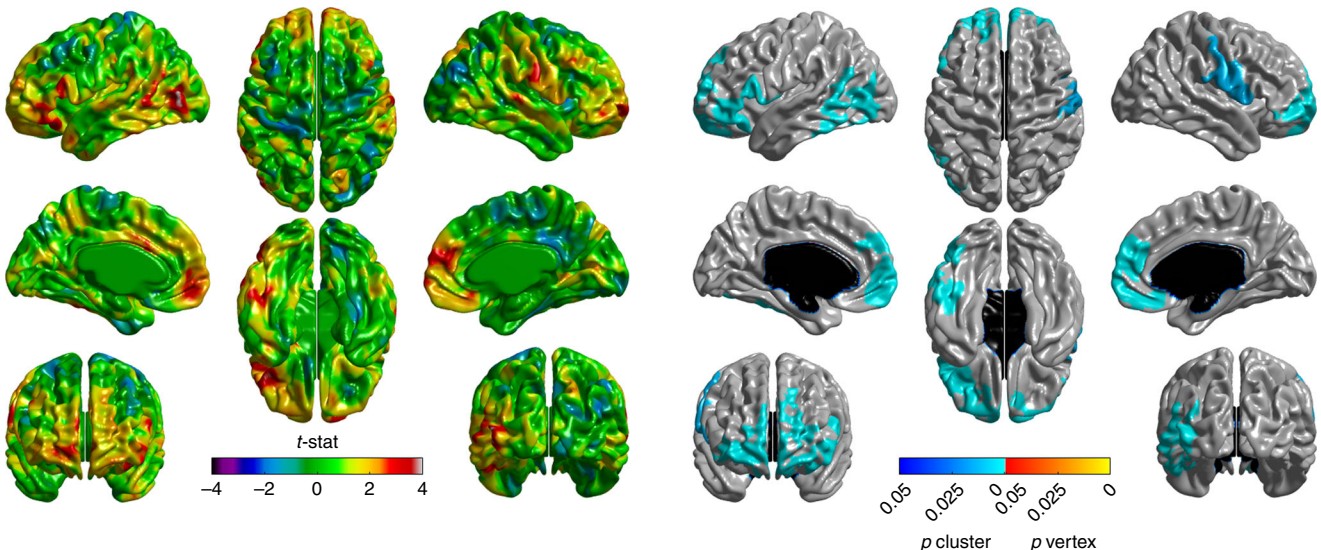

**Fig. 1** Cortical thinning in PD. Changes in cortical thickness in PD patients compared to healthy controls over the 1-year study period. Left panel: *t*-statistic comparing PD to HC at baseline and follow-up [(PD2-PD1)–(HC2-HC1)], at each cortical vertex. Right panel: thresholded map showing areas of significant difference. Cluster-corrected *p*-value in blue

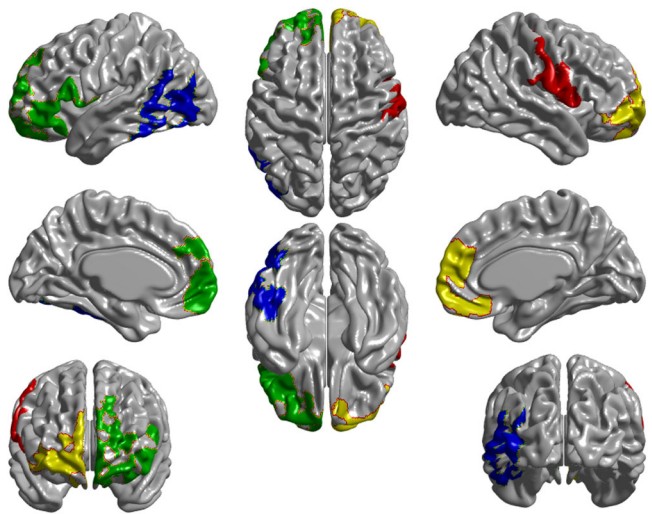

**Fig. 2** Significant clusters of cortical thinning in PD. Four peak clusters of cortical thinning in PD were identified in parts of (1) left occipital lobe (nverts = 1081 [5.53 resels], *p* < 0.0001, (2) left frontal lobe (nverts = 2115 [10.99 resels], *p* < 0.0001), (3) right frontal lobe (nverts = 3720 [15.79 resels], *p* < 0.0001), and (4) right somatomotor cortex (nverts = 1990 [16.56 resels], *p* < 0.0001). nverts: number of vertices in the significant area

and post-central gyrus and supramarginal gyrus (Fig. 2). The frontal lobe atrophy mostly involved the inferior parts of lateral and medial frontal cortex, including the orbitofrontal cortex, frontal pole, ventromedial, and rostral dorsomedial prefrontal cortex. On the lateral surface, inferior and medial frontal gyri were implicated, with extension to the superior frontal gyrus on the left. In order to assess the likely functional consequences of cortical thinning in PD, we assigned each cortical parcel to one of the seven intrinsic brain networks derived from resting-state functionl MRI (fMRI) by Yeo et al.[28]. Cortical regions belonging to the limbic, frontoparietal, and ventral attention networks on average demonstrated the greatest cortical thinning, as well as anterior parts of the default mode network (DMN; Fig. 3). There were no areas showing greater loss of cortical thickness in HC than in PD.

**Connectivity and disease propagation**. The disease propagation hypothesis predicts that cortical brain atrophy at follow-up should depend on connectivity of cortical areas to affected areas at baseline. We had previously shown predominant subcortical atrophy in de novo PD in this population. Connectivity between any cortical area and this putative subcortical "disease reservoir" should therefore predict the degree of cortical thinning relative to age-matched controls when comparing follow-up to baseline. We define the disease reservoir as the subcortical areas showing atrophy in PD patients relative to controls at the initial evaluation, as determined previously using DBM[9]. For the current analysis, the brain was segmented into 463 cortical and subcortical regions of interest[29]. Each surface vertex from the cortical thickness images was mapped to the nearest volumetric parcel (see Methods section). To examine the propagation hypothesis, the connectivity ($\text{Conn}_{ij}$) between any cortical parcel $i$ and any parcel belonging to the disease reservoir $j$ was measured. Here we calculated $\text{Conn}_{ij}$ based on healthy brain connectomes (see Methods section). The analysis was performed for the whole brain and for each hemisphere separately using Spearman's correlation to examine the relation between disease progression and exposure. Disease exposure was defined as the product of connectivity and atrophy measure in the corresponding disease reservoir region $j$ at the onset of the disease, summed over all possible connections.

$$\text{Disease exposure}(i) = \sum_j \text{Conn}_{ij} \cdot \text{Atrophy}(j)$$

Both functional and structural connectivity were used to test the propagation hypothesis (Fig. 4). Connectomes were generated using data from young healthy individuals to calculate the $\text{Conn}_{ij}$. Atrophy($j$) was set equal to the *z*-value at each region $j$ of the independent component demonstrating atrophy in PD compared to HC from the previously computed DBM map from the baseline MRI[9]. Note that for areas not showing significant atrophy at baseline the value of Atrophy($j$) is equal to 0.

First, a functional connectome was generated using resting-state fMRI to define $\text{Conn}_{ij}$. Significant correlation was observed between regional cortical thinning and disease exposure ($r = 0.19$, $p < 0.0001$). Significance was confirmed by permutation testing where we kept the connectivity structure (i.e., subcortical–cortical connections) intact and permuted the cortical thinning values

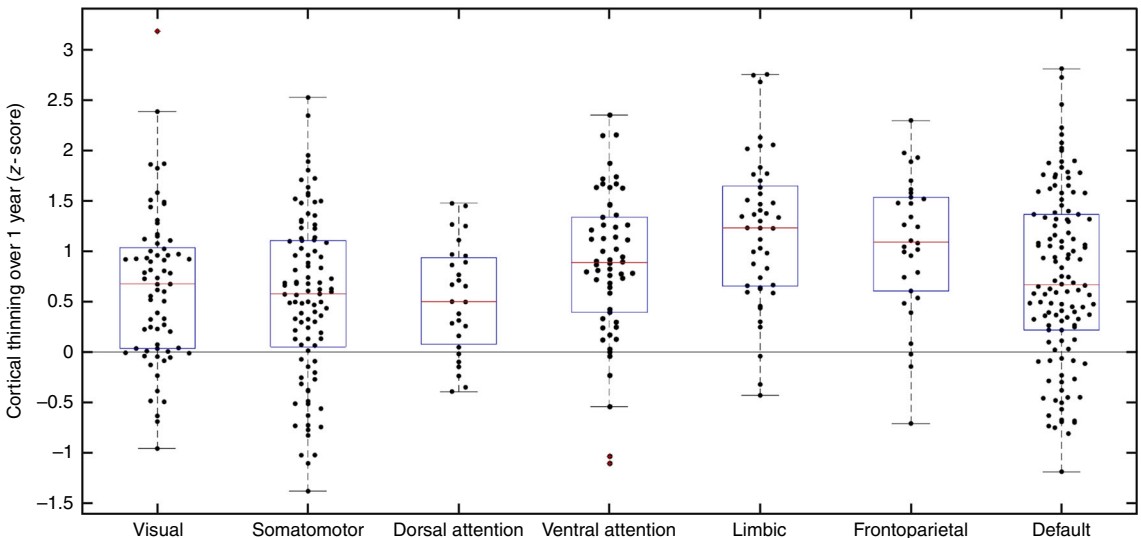

**Fig. 3** Intrinsic networks affected by cortical thinning. Cortical thinning in PD vs HC in the seven intrinsic brain networks of Yeo et al.[27]. All seven intrinsic brain networks demonstrated significant cortical thinning with $p < 0.0001$ (permutation test $n = 10,000$): visual network ($\mu = 0.63$; 95% CI $\in [0.46, 0.81]$), somatomotor network ($\mu = 0.53$; 95% CI $\in [0.37, 0.69]$), dorsal attention network ($\mu = 0.52$; 95% CI $\in [0.32, 0.73]$), ventral attention network ($\mu = 0.86$; 95% CI $\in [0.67, 1.94]$), limbic network ($\mu = 1.2$; 95% CI $\in [0.98, 1.40]$), frontoparietal network ($\mu = 1.05$; 95% CI $\in [0.80, 1.28]$), and default mode network ($\mu = 0.73$; 95% CI $\in [0.57, 0.88]$). + outliers within each network.

($n = 10,000$) and the 95% confidence interval was measured using bootstrapping ($n = 10,000$) ($r \in [0.11, 0.28]$, p_perm $< 0.0001$). Moreover, cortical thickness of both the left ($r = 0.22$, $p = 0.0005$, $r \in [0.09, 0.34]$, p_perm $= 0.0004$) and right hemisphere ($r = 0.27$, $p < 0.0001$, $r \in [0.16, 0.38]$, p_perm $< 0.0001$) significantly correlated with the disease exposure from bilateral affected regions.

Second, the same hypothesis was tested using structural connectivity as measured by diffusion-weighted MRI (DW-MRI). Though the correlation was not significant at the whole-brain level ($r = -0.03$, $p = 0.239$, $r \in [-0.12, 0.06]$, p_perm $= 0.762$), change in cortical thickness and disease exposure within the individual hemispheres were significantly correlated. Similar results were observed within the left hemisphere ($r = 0.14$, $p = 0.016$, $r \in [0.01, 0.27]$, p_perm $= 0.016$) and the right hemisphere ($r = 0.11$, $p = 0.044$, $r \in [-0.02, 0.25]$, p_perm $= 0.041$). Failure to capture the relationship at the whole-brain level may be due to inherent limitations in DW-MRI to detect interhemispheric connectivity[30–33] (See Discussion section).

For comparison, when only considering pure Euclidean distance, we found that cortical thickness of neither the whole brain ($r = -0.06$, $p = 0.104$, $r \in [-0.15, 0.03]$, p_perm $= 0.105$) nor the left hemisphere ($r = -0.09$, $p = 0.088$, $r \in [-0.22, 0.04]$, p_perm $= 0.084$) correlated with the disease exposure from bilateral-affected regions. However, when only considering the right hemisphere, we observed significant results ($r = -0.21$, $p = 0.0007$, $r \in [-0.33, -0.09]$, p_perm $= 0.0006$).

Furthermore, we tested the relative contribution of functional and structural networks to disease spread using a multivariate model for each hemisphere. Within the left hemisphere, the analysis shows that both functional connectivity ($\beta = 0.18$, $t(220) = -3.88$, $p < 0.001$) and structural connectivity ($\beta = 0.04$, $t(220) = -2.27$, $p = 0.02$) significantly relate to cortical atrophy. Within the right hemisphere, again functional connectivity ($\beta = 0.12$, $t(220) = -2.56$, $p = 0.01$) and structural connectivity ($\beta = 0.08$, $t(220) = -2.77$, $p < 0.01$) significantly relate to cortical atrophy. These results suggest that both types of connectivity play a role in the disease-spread model; however, functional connectivity appears to be more significant.

As a follow-up exploratory analysis, we tested whether striatal and non-striatal regions within our disease reservoir may contribute differently to disease exposure. Based on functional

connectivity, analysis restricted to a basal ganglia reservoir showed an increased effect size within the left hemisphere ($r = 0.29$, $p < 0.0001$, $r \in [0.16, 0.40]$, p_perm $< 0.0001$), right hemisphere ($r = 0.37$, $p < 0.0001$, $r \in [0.26, 0.47]$, p_perm $< 0.0001$), and across the whole rain ($r = 0.29$, $p < 0.0001$, $r \in [0.21, 0.37]$, p_perm $< 0.0001$) compared to the analysis using all affected subcortical regions. When repeating this analysis for structural connectivity, the effect size observed for a basal ganglia-restricted analysis within the left hemisphere ($r = 0.14$, $p = 0.016$, $r \in [0.01, 0.27]$, p_perm $= 0.017$), right hemisphere ($r = 0.15$, $p = 0.012$, $r \in [0.01, 0.28]$, p_perm $= 0.011$), and across the whole brain ($r = 0.02$, $p = 0.35$, $r \in [-0.08, 0.11]$, p_perm $= 0.35$) were similar to the analyses considering all subcortical structures.

**Relation to cognition and cerebrospinal fluid (CSF) measures.** Additional analyses restricted to the PD group were performed to investigate the relationship between cortical thinning of the four identified clusters and other disease-related measures (Fig. 5). To test whether CSF measures may contribute to the cortical thinning within each of the clusters, we conducted partial correlations (controlling for age and gender). We found that changes in the left frontal ($r = -0.16$, $p = 0.05$) and left occipital cluster ($r = -0.18$, $p = 0.037$) both correlated negatively with CSF $A\beta_{42}$. Changes of the left occipital cluster ($r = 0.17$, $p = 0.043$) also correlated with CSF $\alpha$-syn. We then performed a linear regression model to examine whether cortical thinning was related to cognitive decline (as assessed by MoCA) and found the overall model, controlling for covariates, was significant ($r^2 = 0.12$, $p = 0.03$). Further post-hoc analysis revealed that only the cortical thinning of the left frontal cluster ($r = -0.19$, $p = 0.03$) significantly related to MoCA changes. Note that these analyses fail to survive correction for multiple comparisons of the four significant clusters.

CSF measures did not correlate with individual time point measures nor changes in MoCA or motor severity as assessed by UPDRS-III (all $p > 0.05$). No significant correlation was observed between cortical thickness of any of the four clusters and measures of UPDRS-III or p-tau$_{181}$. In the control group, no significant correlations between cortical thickness and measures of UPDRS-III, MoCA, or CSF biomarkers were observed.

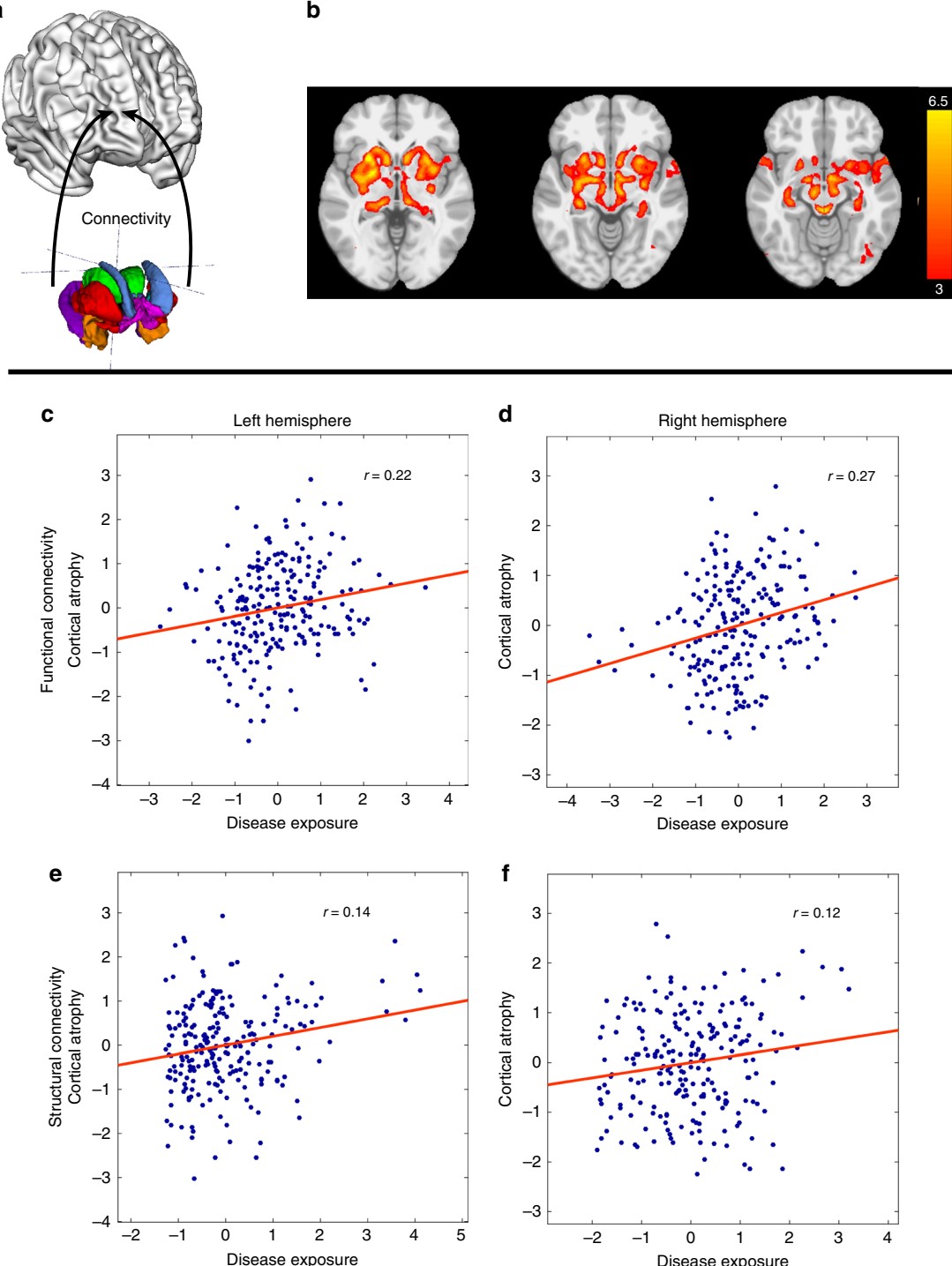

**Fig. 4** Testing the propagation model. **a** Schematic of the propagation model. The model used a disease reservoir (colored regions) consisting of 15 subcortical regions of interest and connectivity from each region to the cortical surface. **b** PD-specific atrophy map used to define disease "content" in each region of interest of the disease reservoir, derived from deformation-based morphometric analysis of patients at baseline (modified from Zeighami et al. 2015). **c**–**e** Correlation between cortical thinning (average $t$-values, PD–HC, $t_2$-$t_1$) of each cortical region and disease exposure (see text for detail). Connectivity based on functional resting state data in the **c** left and **d** right hemisphere as well as connectivity based on diffusion-weighted data in the **e** left and **f** right hemisphere are depicted

## Discussion

The present study investigated 105 PD and 57 control participants over a 1-year period following initial diagnosis. Early PD progression was associated with significant cortical changes. While aging was also associated with cortical thinning, PD patients demonstrated significantly greater reduction in cortical thickness than controls (0.028 mm vs 0.019 mm reduction on average). This was observed mostly in parts of the left occipital and bilateral frontal lobes and right somatomotor-sensory cortex—regions largely belonging to the limbic, frontoparietal, ventral attention, and default mode networks.

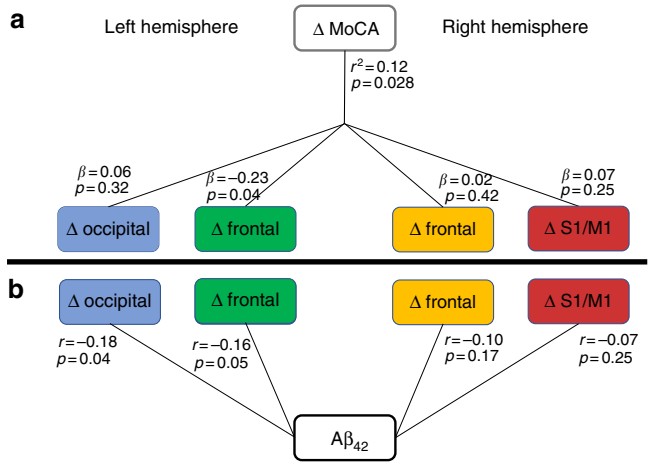

**Fig. 5** Relation to markers of cognitive decline. **a** Linear regression model between changes (delta) in cortical thickness clusters and cognitive score (MoCA). The overall model was significant; specifically, thinning of the left frontal cluster ($\beta = -0.23$, $p = 0.03$) and age ($\beta = 0.22$, $p = 0.01$) were significantly related to decline in cognitive performance. **b** Partial correlations of CSF $A\beta_{42}$ with thinning of each cortical thickness cluster revealed significant correlations in the left hemispheric clusters. All analyses controlled for age and gender

We tested the theory that neurodegeneration in PD results from a disease-spreading process that enters the cerebral cortex from initially affected subcortical areas. Per the network propagation hypothesis, neurodegenerative diseases result from the aggregation and propagation of misfolded proteins, which in turn results in neuronal death and brain atrophy[2,4]. The distribution of focal cortical thinning should therefore be contingent on connectivity to already affected areas, which we call a "disease reservoir" by analogy to the spread of epidemics in populations. In the present study, we tested this hypothesis in PD with longitudinal data. We found that cortical regions with greater connectivity (measured functionally or structurally) to the subcortical disease reservoir identified at the time of diagnosis demonstrated greater cortical atrophy over the 1-year period. This is in line with our previous work using DBM in the initial scan from this dataset[9], which indicated that regions demonstrating the greatest atrophy were those belonging to a connectivity network with an epicenter in substantia nigra—the hypothesized origin of misfolded protein spread to the supratentorial cerebrum in PD.

Some aspects of our model deserve mention. Predictions based on physical distance were similar to topographical distance based on DW-MRI in the right hemisphere. This may be a result of the known distance bias of probabilistic tractography[31] or that the brain is organized such that neuronal connectivity correlates with Euclidean distance[32]. Resting-state connectivity, however, better predicted atrophy than tractography or topographical distance, thus supporting the notion that connectivity improves our measure of propagation above and beyond consideration of geometrical distance alone. Moreover, there is no described mechanism by which a protein spreads physically through the neuropil. While functional connectivity models fit the data at a whole-brain level, we only observed a relationship between connectivity and thinning in individual hemispheres when using a DW-MRI connectome. This is perhaps due to inherent limitations in DW-MRI to capture interhemispheric connections[33]. Finally, the disease exposure analysis limited to the basal ganglia only yielded greater effects on cortical thinning than the whole subcortical reservoir model. This might suggest a greater contribution of basal ganglia to cortical propagation, although our results do not

unequivocally demonstrate this. Mathematical models of the dynamic spread through brain networks, accounting for factors such as protein accumulation and clearance, have been proposed in recent years for AD[34–36]. Such models might be usefully applied to PD.

In our post-hoc analysis, greater reduction in cortical thickness of the left frontal cluster in PD over the 1-year period was associated with worsening cognition (as assessed by the MoCA), although the *p*-value of 0.03 would not survive correction for multiple comparisons. This mirrors results from other studies that have similarly found trending results[22,37]. While dementia is rare in early PD, especially as our cohort was selected to exclude dementia at baseline, we still see greater cortical thinning of frontal clusters in the PD group 1-year postdiagnosis. Cortical neuronal degeneration could be occurring earlier in the clinical course of PD than once thought. Although PD is primarily thought of as a movement disorder, it is recognized to be a brain-wide neurodegenerative process that spreads up from brainstem into cortex, as originally suggested by Braak[38].

In this dataset, only cortical thinning of the left frontal cluster related to cognitive decline. Furthermore, thinning in this frontal cluster was weakly related to reduced $A\beta_{42}$ at baseline (Fig. 5). Lower CSF $A\beta_{42}$ and α-syn was also associated with occipital thinning, which did not correlate with cognitive decline. The frontal clusters demonstrating greater loss of cortical thickness in PD involved lateral and medial frontal cortex, predominantly in ventral and orbitofrontal areas. These regions of frontal lobe belong to the limbic system, the frontoparietal systems implicated in focused attention, and the DMN implicated in memory retrieval. This pattern of frontal atrophy is similar to that described in a subgroup of AD patients identified by hierarchical clustering[39] and in patients with behavioral variant fronto-temporal dementia[1]. A notable difference between the pattern of cortical atrophy described here and that commonly seen in AD is the relative absence of more posterior involvement (precuneus, posterior cingulate, and parietal lobe) in our PD cohort. The effect seen here is much more anterior, possibly reflecting the propagation of disease via dopaminergic or frontostriatal projections. Nonetheless, the DMN is also disrupted in AD[40,41] and a target of amyloid plaques[42]. The anterior DMN may be an avenue through which dementia develops in PD. We note also that, in the Braak model of AD, the initial cortical site of amyloid-β involvement is the ventral frontal cortex[43] and that the observed ventral frontal atrophy pattern in PD shows a high degree of similarity to early $A\beta_{42}$ deposition measured by positron emission tomography in AD[44].

While cognitive impairment is a recognized consequence of PD[23], its underlying mechanism is not well understood. Our findings suggest that, in the PD group, higher $A\beta_{42}$ deposition in the brain at baseline (indicated by lower CSF $A\beta_{42}$ levels[45]) predicted greater cortical thinning of the left frontal cluster over the 1-year period. Though our results are not conclusive, they suggest that $A\beta_{42}$ may make the PD brain more susceptible to cortical thinning and, in turn, more prone to cognitive decline. Indeed, recent positron emission tomographic studies have revealed that cortical amyloid deposits are associated, in a dose-dependent manner, with increased risk for cognitive decline in PD[46]. Additionally, low CSF $A\beta_{42}$ has previously been associated with the risk of subsequent dementia in PD[47]. These findings provide evidence in favor of potentially overlapping mechanisms for dementia in PD and AD. Previous research found that 28% of PD patients have AD pathological features at postmortem[48]. While it can be argued that the findings observed in the present study may be driven by coincidental occurrence of AD, we found that only 5 of the 105 PD subjects met a proposed CSF amyloid-positive criterion[49]. A possible alternate explanation is that

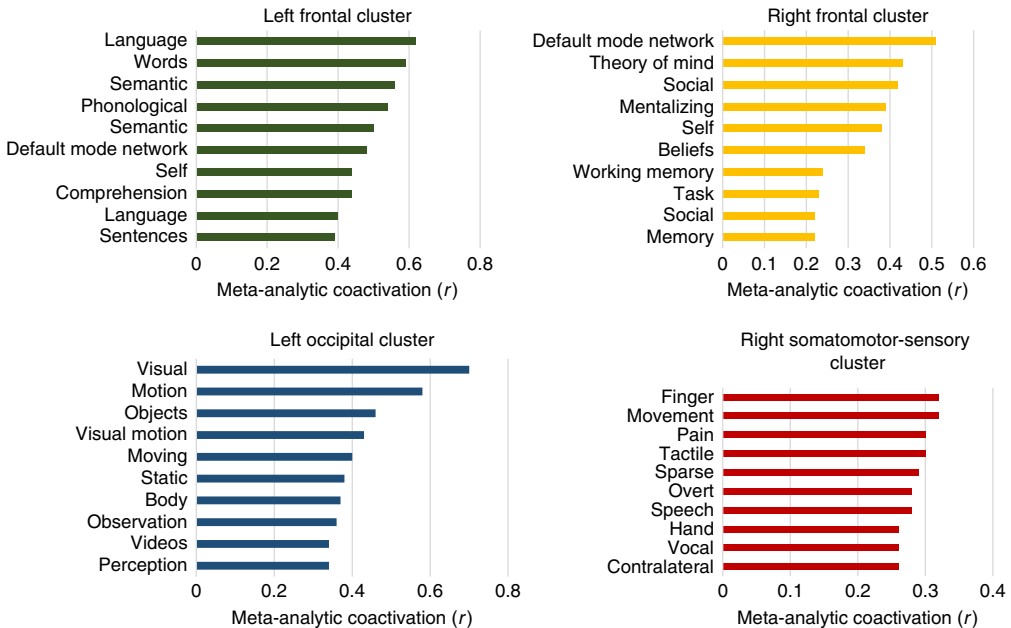

**Fig. 6** Functional anatomy of cortical thinning in PD. Peaks within each cluster were associated with expressions used in publications using Neurosynth. The *x* axes represent correlation values between the cluster and each concept's meta-analytic map generated using seed-based networks. The *y* axes represent the most highly correlated concepts per cluster

synergy between Aβ$_{42}$ and α-syn deposition promotes cognitive impairment in PD[50]. Aβ$_{42}$ has been found to promote gray matter atrophy in distributed regions of the parietal and frontal lobe in the normal aging brain[51], akin to the atrophy pattern observed in the present study; α-syn may accelerate the neurodegenerative process associated with normal age-related Aβ$_{42}$ deposition, especially in regions connected to the PD disease reservoir.

The pattern of cortical atrophy in these de novo PD patients may account for the constellation of mood and cognitive deficits that eventually characterize PD dementia. Involvement of the lateral prefrontal cortex and striatum could lead to executive dysfunction. Inferior frontal areas also affected in behavioral variant frontotemporal dementia may be implicated in apathy, compulsive behaviors, changes in personality, and social deficits. Finally, DMN involvement could account for Alzheimer-like memory deficits while degeneration of ventral visual stream areas could explain visual hallucinations. Using the Neurosynth program and database[52], we searched for the terms most often associated in fMRI publications with each anatomical cluster demonstrating atrophy in PD (Fig. 6). The left occipital–temporal cluster is associated with (visual) motion, observation, perception, and vision. The right somatomotor-sensory cluster is associated with movement. The frontal clusters are associated with expressions related to language and to the terms theory of mind, self, social, mentalizing, and memory. The atrophic regions observed here subserve the cognitive functions that fail in PD dementia.

Findings from this study should be considered in light of some limitations. While the cellular events leading to changes in cortical thickness are not clear, cortical thickness as measured with T1-weighted MRI is nonetheless believed to reflect neuronal density along with glial support and dendritic arborization[53,54]. In PD, reductions in cortical thickness most likely reflect neurodegeneration. In the current paper, we address the "protein-spread" model, which we note remains controversial and lacking in supportive human evidence[6,7]. With our current dataset and imaging modalities, we cannot rule out the possibility of a stochastic attack on the brain as a whole that would give the appearance of network spread. Areas that are connected could

share features (e.g., gene expression[55]) that make them similarly vulnerable to a specific pathophysiological process. In this case, the map of atrophy would look like a network but not be driven by connectivity. Additionally, neurodegeneration could propagate due to loss of trophic factors having nothing to do with misfolded protein spread. Moreover, we cannot measure α-syn accumulation but only its consequence (cortical thinning), and the two may have a complex and nonlinear relationship. However, the present finding that a connectivity model explains disease progression better than chance lends support to the network-propagation hypothesis. The inclusion criterion of MoCA > 26 may have biased both the PD and HC pool. Also, while our sample is relatively large, the PPMI sample is a research-based cohort and may not be truly representative of a population-based cohort. Finally, most patients were prescribed dopaminergic treatments after diagnosis, which may mask the magnitude of cognitive decline and motor symptoms and their relationship with brain atrophy. Future longitudinal studies and follow-up with patients at later stages of the disease as well as postmortem confirmation are needed to understand the relationship between cortical atrophy and the underlying disease process.

The current study is the largest to date to assess longitudinal measures of cortical thickness using 3 T MRI in a de novo PD cohort. Our findings suggest that disease propagation to the cortex occurs earlier than previously thought. Focal cortical thinning may be contingent on connectivity to a disease reservoir: the greater the connections, the higher the disease exposure, and the more the cortical atrophy. PD progression involves subcortical α-syn spreading through intrinsic brain networks to the cortex, possibly in synergy with Aβ$_{42}$, which may be the harbinger of cognitive decline.

## Methods

The PD patients included in this study met the PPMI inclusion criteria[8] consisting of asymmetric resting tremor or asymmetric bradykinesia or any two of bradykinesia, resting tremor, and limb rigidity. Patients had to have been diagnosed within 2 years of enrollment and not have received anti-parkinsonian medications. Diagnosis was confirmed with dopamine transporter imaging using single-photon emission computed tomography. Control subjects were free of neurological disease.

Each participating PPMI site received approval from their local institutional review board and obtained written informed consent from all subjects. Analysis of this dataset was approved by the Montreal Neurological Institute Research Ethics Board. Only participants with 3 T MRI data at both initial visit and 1-year follow-up were included in the present study. Of the 237 PD patients with 3 T acquisition at baseline, 122 had MRI data at the 1-year follow-up. For controls, 118 subjects had 3 T acquisition at baseline, and 62 had MRI data at the 1-year follow-up.

All non-categorical data were normally distributed as per Shapiro–Wilk tests. Group difference in sociodemographic and neuropsychological variables were analyzed using the $\chi^2$ test for categorical data and $t$-test for normally distributed data.

**MRI acquisition**. T1-weighted MRI scans were acquired in the sagittal plane on 3 T scanners at each study site using a magnetization-prepared rapid-acquisition gradient echo sequence (full study protocol: http://www.ppmi-info.org/study-design/research-documents-and-sops/). Acquisition parameters were as follows: repetition time = 2300/1900 ms; echo time = 2.98/2.96/2.27/2.48/2.52 ms; inversion time = 900 ms; flip angle: 9°; 256 × 256 matrix; and $1 \times 1 \times 1$ mm$^3$ isotropic voxel. Acquisition site was used as a covariate in all analyses.

**Cortical thickness**. Cortical thickness models were generated from the T1-weighted 3 T MRI scans using the CIVET2.1 preprocessing pipeline[56]. A summary of the steps involved follows; full documentation can be found online (http://www.bic.mni.mcgill.ca/ServicesSoftware/CIVET). The T1-weighted MRIs were linearly registered to the MNI-ICBM152 volumetric template[57,58]. Images were then corrected for signal intensity non-uniformity[59], and a brain mask was calculated from each input image. Images were then segmented into gray matter, white matter, CSF, and background[56]. After the CIVET pipeline, quality control was carried out by two independent reviewers using previously described criteria[60] to ensure adequate quality of the T1-weighted volume images, such as linear and nonlinear registration, and to exclude distortions in gray and white matter surfaces and motion artifacts. Seventeen of the 122 PD and 5 of the 62 HC failed quality control.

The distance between the corresponding vertices of inner and outer cortical surfaces were evaluated by a fully automated method (Constrained Laplacian Anatomic Segmentation using Proximity algorithm[61]) to provide a measure of cortical thickness at each vertex. The white and gray matter interface was fitted across 40,692 vertices in each hemisphere to create the inner cortical surface, then expanded to fit the gray matter and CSF interface to create the outer cortical surface. The surfaces of each subject were nonlinearly registered to an average surface created from the volumetric ICBM152 template. In order to test the propagation hypothesis, the brain was segmented into 448 cortical and 15 subcortical regions[29], and each of the 40,692 cortical vertices from the cortical thickness analysis was interpolated and assigned to one of the 448 cortical parcels.

Changes in cortical thickness were calculated by subtracting the values ($\Delta t = t_1 - t_2$) at the 1-year follow-up ($t_2$) from the baseline ($t_1$) for both groups using the SurfStat software package (http://www.math.mcgill.ca/keith/surfstat/). Difference between PD and controls [(PD$_1$-PD$_2$)–(HC$_1$-HC$_2$)] were analyzed statistically based on random field theory with a threshold of $p < 0.05$ corrected for multiple comparisons over the entire cortical surface[62].

In order to address the relationship between cognitive decline and cortical thinning over the 1-year period, we performed a linear regression model. Change in MoCA was the dependent variable, while mean cortical thinning values from each of the four clusters distinguishing PD from controls were the independent variables. Gender and age at baseline were used as covariates. Observed differences in cortical thickness were then correlated to other potential disease-related measures (i.e., UPDRS-III, MoCA, CSF α-syn, and tau and Aβ$_{42}$ markers) using Pearson partial correlations as post-hoc analysis. All the analyses were performed using MatLab 2015b statistics toolbox. Within PD patients, the difference in cognitive performance over the 1-year interval was examined using a linear regression model with MoCA as the dependent variable and the thinning of the four cortical clusters as independent variables.

To test whether cortical thickness of a specific cluster was linked to cognitive performance, post-hoc analyses using partial correlations were conducted. Age at baseline and gender were used as confounding covariates for all analyses including for partial correlations, linear regression model, and for obtaining cortical thickness.

**Deformation-based morphometry**. Data were analyzed from the baseline visits of newly diagnosed PD patients ($n = 232$) and an age-matched control group ($n = 117$) obtained from the PPMI database. All subjects' initial visit's 3 T high-resolution T1-weighted MRI scans underwent preprocessing steps including denoising[63], intensity non-uniformity correction[59], and intensity range normalization. Next, each subject's T1-weighted image was first linearly, and then nonlinearly, registered to the MNI-ICBM152 template. Using the resulting nonlinear transformation fields, we calculated deformation morphometry maps (i.e., the determinant of the Jacobian matrix of transformation). DBM maps were then concatenated and independent component analysis (ICA) was employed to define independent sources of deformation in the brain. For the resulting ICA components ($n = 30$), the PD group was compared to healthy controls (unpaired $t$-test). The atrophy level significantly differed between PD and control groups in only one of these components after correcting for multiple comparisons (Bonferroni correction),

suggesting that this component captures PD-specific alterations in the brain. We used this component (referred to as PD-ICA here) as our atrophy map. The component score (atrophy measure) in PD-ICA for each subject was significantly correlated with disease severity as measured by UPDRS-III and single-photon emission computed tomography striatum binding ratio. Please see Zeighami et al.[9] for further details.

**Connectivity and disease propagation**. The difference in cortical thinning in PD vs controls was used to define disease progression. DBM on the baseline MRI was employed to assess disease-related brain atrophy and to define a source, or reservoir, for disease propagation. Based on a parcellation that is a subdivision of the common FreeSurfer implementation of the Desikan–Killiany Atlas, gray matter was segmented into 463 regions of interest of approximately similar size[29]. Of these parcels, 448 were cortical and 15 were subcortical (thalamus, caudate, putamen, pallidum, accumbens, hippocampus, amygdala in each hemisphere and brainstem). Each surface vertex from the cortical thickness analysis was then mapped to the corresponding nearest whole-brain parcel from the Atlas. The "disease reservoir" was defined as the 15 subcortical parcels from the Atlas, and the Atrophy value at each parcel was obtained from the DBM map.

To generate brain networks and connectomes, data-driven brain parcellations from young healthy individuals were used[29]. The functional connectivity map was derived from resting-state fMRI data of 40 young healthy subjects (25.3 ± 4.9 years, 24 males)[64]. The scans were done using Siemens Medical Trio 3 T MRI scanner, consisting of a high-resolution T1-weighted image, as well as T2*-weighted images with blood-oxygen-level-dependent contrast (3.3 mm$^3$ isotropic voxels, TR 1.9 s). The dataset was preprocessed using conventional steps consisting of rigid body motion correction, correcting for nuisance variables (white matter, CSF, motion parameters), and low pass filtering. The structural connectivity map was generated using the Illinois Institute of Technology Human Brain Atlas v.3 (freely available at http://www.iit.edu/~mri/Home.html). The atlas was constructed using data from 72 neurologically healthy subjects (20–40 years, 28.8 ± 5.5 years, 30 males). All subjects were scanned using a 3 T General Electric MRI scanner (GE, Waukesha, WI) with the following parameters (TR = 5800 ms, TE = 94ms, voxel size = 3 mm$^3$, 256 × 256 final image matrix, FoV = 24 × 24 cm$^2$, $b$ = 900 s mm$^{-2}$, 45 slices for 12 diffusion directions uniformly distributed in 3D space, scan time: 21 min 57 s)[65]. Imposing a maximum of 500 mm trace length and a curvature threshold of ±90, the anatomical connection probability was calculated between all pairs of regions using a probabilistic graph-based tractography algorithm[66].

Spearman's correlation was used to investigate the relationship between disease progression in cortical areas (cortical thinning) and the 'disease exposure' for each region in the PD-ICA (defined by the average values of each region weighted by connectivity). The same procedure was performed to test propagation hypotheses along the functional and structural connectomes. The propagation analysis was performed for the whole brain and for each hemisphere separately.

**CSF measures**. CSF Aβ$_{42}$, α-syn, t-tau, and p-tau$_{181}$ were collected by lumbar puncture at PPMI clinical sites[67]. Frozen aliquots (at −80 °C) were then shipped and assessed using the MAP Luminex platform (Luminex Corp, Austin, TX) with Innogenetics (INNO-BIA AlzBio3; Ghent, Belgium; for research use-only reagents) immunoassays used for Aβ$_{42}$, t-tau and p-tau$_{181}$. CSF α-syn assay was performed at Covance using a commercially available ELISA Assay Kit (Covance, Dedham, MA)[68]. To evaluate the possible contamination of blood in CSF—a factor thought to influence the level of some proteins including α-syn and Aβ$_{42}$[69,70]—the relationship between CSF hemoglobin and other CSF measures was assessed. No significant association was observed (all $p < 0.05$), suggesting that there is no effect of added blood on concentration of CSF measures. Owing to the ongoing nature of the study and incomplete data, only CSF data from baseline were used in this analysis[68].

**Code availability**. All code written by the authors of this manuscript is available upon request.

**Data availability**. The DBM maps are available at https://neurovault.org/collections/860/. Cortical thickness maps will be made available on request. The CIVET pipeline is available at http://www.bic.mni.mcgill.ca/ServicesSoftware/CIVET. The functional and structural connectome data are from cited sources and available upon request from the authors of the related publications.

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

## Acknowledgements

PPMI—a public–private partnership—is funded by the Michael J. Fox Foundation for Parkinson's Research and funding partners, including AbbVie, Avid, Biogen, Bristol-Myers Squibb, Covance, GE Healthcare, Genentech, GlaxoSmithKline, Lilly, Lundbeck, Merck, Meso Scale Discovery, Pfizer, Piramal, Roche, Sanofi Genzyme, Servier, Teva, and UCB. This research was supported by grants from the Michael J. Fox Foundation for Parkinson's Research, the W. Garfield Weston Foundation, and the Alzheimer's Association; the Canadian Institutes of Health Research; and the Natural Sciences and Engineering Research Council of Canada to A.D. Y.Y. is a Vanier Scholar and receives funding from the Canadian Institute of Health Research. Y.Z. holds the Jeanne Timmins Costello Fellowship.

## Author contributions

Y.Y., Y.Z., and A.D. contributed to the study design and interpretation. Y.Y., Y.Z., T.B., K.L., U.V., and M.D. analyzed the data. V.F., B.M., D.L.C., P.H., and A.G. contributed to methods. Y.Y. drafted the initial manuscript; all authors contributed to writing of this manuscript.

## Additional information

**Competing interests:** The authors declare no competing financial interests.

