## [Peer Review File · Nature Communications]

Reviewers' comments:

Reviewer #1 (Remarks to the Author):

This paper addresses the interesting and important issue of network-based neurodegeneration in a disease that remains relatively poorly defined from this perspective. The authors present the case that Parkinson's disease may propagate longitudinally according to functional and structural connectivity between cortical regions and a predominantly subcortical 'disease reservoir'.

Although the concept is attractive, I have two major reservations about the data presented.

The first concerns the formulation of the disease reservoir and 'disease exposure' and how the latter is used to predict cortical atrophy. The authors report correlations between particular cortical regions and 'disease exposure' where exposure is defined (line 177) as a product of connectivity and atrophy in connected disease reservoir regions. The inclusion of this atrophy product term I believe entails bias; atrophy in the index cortical region might tend to correlate with atrophy in the connected region, independently of the strength of connectivity between the regions. This atrophy correlation could for example be based on anatomical proximity, regional geometry or some thresholding effect of overall disease burden, rather than any network-specific attribute. A more principled approach to assess the relationship to connectivity per se would employ a two stage process: i) assess the atrophy correlation between any pair of cortical regions, then ii) assess if the strength of this correlation itself correlates with the connectivity between the regions. It is not obvious that these approaches are mathematically equivalent.

A more basic issue concerns the interpretation of the correlation. Even if a relationship to intrinsic connectivity is unambiguously established, this does not establish (as the authors argue) that disease propagation follows connectivity. A connected system of neurons could alternatively be subject to a stochastic attack that selectively targets the system as a whole. A number of previous studies (notably in frontotemporal dementia) have related disease atrophy patterns to the intrinsic functional connectome determined from the healthy brain; in this respect, the present study is not particularly innovative. However, none of these studies resolves the key issue of propagation. The most direct way to address this would be to measure inter-regional functional connectivity changes directly in a patient cohort, then show that these disease-related connectivity changes predict subsequent atrophy in those specific regions (but not other connected regions). The present study (like its predecessors) does not examine this mechanism directly.

I have a less substantive concern around the interpretation of the CSF beta-amyloid correlation with cortical thinning; what is the basis for this? One possibility (which the authors acknowledge) is that this was driven by coincident Alzheimer pathology. It is not clear how many patients in the Parkinson's cohort might have met CSF criteria for underlying Alzheimer pathology but at the very least this possibility tends to confound conclusions about the effect of Parkinson's disease per se.

Reviewer #2 (Remarks to the Author):

This paper's major claim is that the atrophy pattern observed within the first year of PD diagnosis is correlated with their novel disease exposure metric (a sum of the atrophy in connected regions, weighted by the functional or structural connectivity strengths). They also perform path modeling to tease apart the relationship of CSF biomarkers, atrophy and measures of cognitive decline. I believe this is the first study of the effect of connectivity on the spatial pattern of disease in PD. I applaud the premise of the paper and believe that the results, if indeed true, are interesting and bring PD into the same fold as other dementias that have been shown to be prion-like. The analysis is generally well done, except for the apparent lack of multiple comparisons corrections.

1) Controlling for multiple comparisons is especially important here, as many of the reported significant p-values are close to (or exceed) the threshold of 0.05 (some tests used a threshold of 0.1 for unclear reasons). My concern is that the results may not survive the level of significance if MC-corrected. In particular, there are many correlations being performed in the analysis presented on page 10 (correlation of MOCA/CSF/regional cortical thinning measures). In addition, the uncorrected p-values for the beta coefficients in the path model are either close to the 0.05 level of significance (or above it).

2) I think it would also be interesting to investigate the relationship between disease burden and atrophy as measured using both structural and functional connectivity, as it may be the case that PD spread depends on both networks. It would be interesting to see a multivariate model incorporating the disease exposure based on both functional connectivity and structural connectivity predicting atrophy. That way, we could compare the relative contribution of both networks to disease spread. The authors may also consider adding demographics to such a multivariate model to see if age/gender has any impact on the correlations observed in Figure 5.

3) It is unclear what exactly is being done in the correlation between disease exposure and atrophy – what is the difference in the whole brain correlations versus the right and left hemisphere only? Are the disease burden values being calculated using only regions in the left hemisphere (i.e. if there is a connected region in the right hemisphere it is ignored)? If the latter is the case, why would the problems with tractography not picking up inter-hemispheric connections result in a decrease of correlation in the whole brain model? In the worst case scenario, we wouldn't have any connections between right and left hemispheres and would be left with the values that are in the separate hemisphere analysis. Please provide more detail.

4) In Figure 5 c and d, it appears that the disease exposure value is not Gaussian (nor should it be, since it depends on structural connectivity), so Spearman's correlation is probably more appropriate here. It also appears that the correlation is being driven by a handful of regions that are highly connected to the subcortical areas – which are these?

5) From the color scale on Figure 5, it appears that cortical atrophy would be correlated with

just subcortical connectivity, not connectivity modulated by atrophy? Does the atrophy of the subcortical region increase the correlation?

6) Finally, the title of the paper is misleading. From what I can tell, no associations were found between network-based metrics and cognitive decline. Associations were found between cortical thinning and MOCA scores, but this had nothing to do with network connectivity. Actually, this would be interesting if you also did a path model that not only looked at change in cortical thickness in the four clusters, but change in cortical thickness and that region's overall connectivity to the subcortical structures.

7) There is a relative lack of description of the structural connectivity. I realize that the data being used has been published, but it may make sense to provide the bare minimum description of SC network construction (is it number of streamlines? What was the number of directions/resolution of the dMRI? What tractography algorithm was used?)

Reviewer #3 (Remarks to the Author):

This manuscript attempts to examine the longitudinal cortical thinning patterns in PD and test its relationship with cognitive functions and network spread hypothesis. Although it is highly motivated, the work has several issues in methods, literature, and interpretations, which reduce the confidence of its conclusion.

1. Abstract: the authors wrote that 'The atrophy pattern in the frontal lobes resembled that described in Alzheimer's Disease'. I am concerned about this statement. Frontal lobes are not the typical regions atrophied in AD. In addition, it is written in the last sentence 'cognitive impairments occur earlier than previously thought in PD'. It is unclear which evidence in the manuscript support this claim.

2. Some key work in network spread hypothesis in the neuroimaging fields should be cited (e.g. Raj Neuron 2012, Zhou Neuron 2012, Iturria-Medina et al, PLOS Computational Biology, 2014).

3. I may not fully agree with the comment that 'cortical thickness analysis may be better able to delineate cortical morphometrical patterns and more sensitive to age or disease-related regional grey matter changes than VBM'. Each method could pick up different patterns of change in disease, depending on the sample, disease etc.

4. In Figure 1, what is the voxel and cluster level threshold for regional cortical thinning comparison between PD and HC? Did the authors check the opposite direction, i.e., (HC1-HC2)-(PD1-PD2) at the same threshold? What are the covariates included in the statistical analysis?

5. It is unclear to me that whether Fig. 3 is derived from the Yeo's parcellation directly or some clusters showing significant group difference. There is no mention of the rationale of such analysis and its relationship to results presented in Figure 1 & 2.

6. My major concern is regarding the analysis between CSF, cortical thickness, and MoCA scores. Some p-values are not even significant or only marginal ($p < 0.1$). Because CSF does not relate to MoCA in the first place, it is hard to interpret the path model of CSF-cortical

thinking-MoCA here and thus the conclusion of synergistic effect with Ab42 is questionable.

7. Lastly, the network spread hypothesis is only weakly supported here. The r-values between disease exposure and atrophy is only 0.15-0.25. As the authors examined cortical thickness, there is no way to evaluate subcortical atrophy (i.e., substantia nigra) changes over time. It is actually important to characterise the longitudinal changes of this so-called disease onset region or reservoir first and then compare it with the rest of the region. VBM approach may be a good alternative to validate the findings here.

8. Also, there is no figure on the disease reservoir. Structural connectivity and functional connectivity preprocessing and connectivity definitions should be provided in details to allow better evaluation of the results. Given the image resolution, the structural and functional connectivity to such small subcortical region (substantia nigra) might be challenging and not sufficiently reliable.

Reviewers' comments:

We thank the reviewers for their kind remarks. We have addressed the comments in detail below. The main change is that we have rearranged the results to emphasize the novel finding here: that cortical thinning in early PD is explained by connectivity to affected subcortical regions. This supports the neuronal spread hypothesis first proposed by Braak. Second, we see if these cortical thickness changes relate to cognitive decline, which they do. We have clearly emphasized results that are statistically significant when corrected for multiple comparisons, and still mention those that are not but are nonetheless interesting. We have emphasized the pertinence and novelty of the findings, notably by referring to recent reviews that cast doubt on the network spread mechanism in PD.

Reviewer #1 (Remarks to the Author):

This paper addresses the interesting and important issue of network-based neurodegeneration in a disease that remains relatively poorly defined from this perspective. The authors present the case that Parkinson's disease may propagate longitudinally according to functional and structural connectivity between cortical regions and a predominantly subcortical 'disease reservoir'.

Although the concept is attractive, I have two major reservations about the data presented.

1) The first concerns the formulation of the disease reservoir and 'disease exposure' and how the latter is used to predict cortical atrophy. The authors report correlations between particular cortical regions and 'disease exposure' where exposure is defined (line 177) as a product of connectivity and atrophy in connected disease reservoir regions. The inclusion of this atrophy product term I believe entails bias; atrophy in the index cortical region might tend to correlate with atrophy in the connected region, independently of the strength of connectivity between the regions. This atrophy correlation could for example be based on anatomical proximity, regional geometry or some thresholding effect of overall disease burden, rather than any network-specific attribute.

We tested for an effect of physical distance using two separate models; one for connectivity (structural or functional) and another for distance. We found an effect of distance in the right hemisphere, but not the left or the entire brain. Disease exposure based on functional connectivity was a better modulator of cortical thinning in all comparisons. (Functional connectivity was also superior to tractography-based structural; connectivity). The statistics are summarized in Table Sup1 below. We have added the following text to the methods: "*When only considering pure Euclidean distance, we found that cortical thickness of neither the whole-brain ($r=-0.06$, $p=0.104$, $r \in [-0.15,0.03]$, $p_{perm}=0.105$) nor left*

hemisphere ($r=-0.09$, $p=0.088$, $r \in [-0.22, 0.04]$, $p_{perm}=0.085$) correlated with the disease exposure from bilateral affected regions. However, when only considering the right hemisphere, we observed significant results ($r=-0.21$, $p=0.0007$, $r \in [-0.33, -0.09]$, $p_{perm}=0.0006$).” In sum, both structural and functional connectomes were better models of disease exposure than physical distance. Nonetheless, there was an effect of distance in the right hemisphere only. A potential reason for this is that the brain is an economically organized spatially embedded system in which regions that are close to each other tend to be interconnected (Ercsey-Ravasz *et al.*, 2013). i.e. distance tends to correlate with connectivity.

We have added the following text to the discussion addressing this issue: “*Some aspects of our model deserve mention. Predictions based on physical distance were similar to topographical distance based on DW-MRI in the right hemisphere. This may be a result of the known distance bias of probabilistic tractography (Jbabdi et al., 2015), or that the brain is organized such that neuronal connectivity correlates with Euclidean distance (Ercsey-Ravasz et al., 2013). Resting state connectivity, however, better predicted atrophy in comparison to both tractography and topographical distance, thus supporting the notion that connectivity improves our measure of propagation above and beyond consideration of geometrical distance alone. Moreover, there is no described mechanism by which a protein spreads physically through the neuropil.*”

With respect to a potential thresholding effect of disease burden, we agree with the reviewer that other variables may make the disease progression appear to be contingent on connectivity, but be in fact determined by other factors. We have added the following text as a limitation: “*Areas that are connected could share features (e.g., gene expression) (Richiardi et al., 2015) that make them vulnerable to a specific pathophysiological process. In this case, the map of atrophy would look like a network but not be driven by connectivity. Additionally, neurodegeneration could propagate due to loss of trophic factors having nothing to do with misfolded protein spread.*”

Table Sup1. Summary table of Spearman correlations between disease exposure and progression.

	r-value	CI 95% lower	CI 95% upper	p-value	p-perm
Disease Exposed based on Functional Connectivity					
bilateral	0.19	0.11	0.28	0.0000	0.0000
right hemisphere	0.27	0.16	0.37	0.0000	0.0000
left hemisphere	0.22	0.09	0.33	0.0005	0.0004
Disease Exposed based on Structural Connectivity					
bilateral	-0.03	-0.13	0.06	0.2388	0.7515
right hemisphere	0.12	-0.02	0.25	0.0411	0.0428
left hemisphere	0.14	0.00	0.28	0.0165	0.0151
Euclidean Distance					

bilateral	-0.06	-0.15	0.03	0.1036	0.1048
right hemisphere	-0.21	-0.33	-0.09	0.0007	0.0006
left hemisphere	-0.09	-0.22	0.04	0.0883	0.0847

2) A more principled approach to assess the relationship to connectivity per se would employ a two stage process: i) assess the atrophy correlation between any pair of cortical regions, then ii) assess if the strength of this correlation itself correlates with the connectivity between the regions. It is not obvious that these approaches are mathematically equivalent.

The reviewer suggests that we correlate two covariance matrices, one which shows the cortico-cortical structural covariance and another which shows the cortico-cortical connectivity (from functional or diffusion data). However, it is already known that these two measures correlate in the normal brain (Gong *et al.*, 2012), and in the Alzheimer Disease brain as well, from the work of Seeley *et al.* (2009). Showing that cortical thickness covariance is predicted by functional or structural connectivity would not address the causality of atrophy, as we did in this longitudinal study. Moreover, here we wanted to test the Braak model in which disease spreads from subcortical to cortical areas, which the reviewer's suggestion would not assess.

3) A more basic issue concerns the interpretation of the correlation. Even if a relationship to intrinsic connectivity is unambiguously established, this does not establish (as the authors argue) that disease propagation follows connectivity. A connected system of neurons could alternatively be subject to a stochastic attack that selectively targets the system as a whole.

We agree with the reviewer that there may be a “stochastic attack” across the system and that we cannot rule out this possibility based on the present study. More generally, showing that connectivity predicts the progression of neuronal loss does not prove the spread hypothesis; it is however one of many levels of analysis that support it (along with e.g. animal experiments).

We have added the following additional text to the Discussion addressing this issue:

“In the current paper, we address the “protein spread” model, which we note remains controversial and lacking in supportive human evidence (Walsh & Selkoe, 2016; Engelender & Isacson, 2017; Surmeier et al., 2017). With our current dataset and imaging modalities, we cannot rule out the possibility of a stochastic attack on the brain as a whole that would give the appearance of network spread...However, the present finding that a connectivity model explains disease progression better than chance lends support to the network propagation hypothesis.”

4) A number of previous studies (notably in frontotemporal dementia) have related disease atrophy patterns to the intrinsic functional connectome determined from the healthy brain; in this respect, the present study is not particularly innovative.

We agree that other studies have tested the role of the connectome in determining patterns of neurodegeneration in dementias such as AD and FTD, and we cite these papers. The novelty of the current work is that (1) it refers to a different clinical population (i.e., PD) with a different misfolded protein (i.e., alpha synuclein), (2) it uses two time-points to measure change in atrophy over time, and (3) uses both functional and diffusion-based connectomes. Almost all previous work has been based on imaging data from a single time point. We hope that #1 and #2 make this work innovative, especially given recent reviews mentioning the lack of human evidence in support of the propagation model in PD (Engelender & Isacson, 2017; Surmeier et al., 2017).

5) However, none of these studies resolves the key issue of propagation. The most direct way to address this would be to measure inter-regional functional connectivity changes directly in a patient cohort, then show that these disease-related connectivity changes predict subsequent atrophy in those specific regions (but not other connected regions). The present study (like its predecessors) does not examine this mechanism directly.

First, our current dataset, namely the Parkinson's Progression Marker Initiative (PPMI), does not have longitudinal resting-state data. We therefore cannot address the reviewer's suggestion of investigating inter-regional functional connectivity changes within PD subjects. However, we don't think this is the best way to test the propagation model. All studies of the propagation hypothesis (in the dementias) published to date (e.g. from Seeley, Raj, Itturia-Medina) used normal connectomes to test the model. This is crucial. The theory is that a normal brain is rendered atrophic through spread of an agent via normal connection pathways. It is not at all clear how one would incorporate individual changes in connectivity in such a model, even if they were available. Second, generating a base connectome from a PD population might allow for better fits, but it is unfeasible. It is known that both PD medications and loss of dopamine affect functional connectivity (e.g., Simioni *et al.*, 2016; Akram *et al.*, 2017), therefore constructing a PD connectome for the purpose of mapping disease propagation would be prone to error (for example, in PD functional connectivity varies considerably on vs off l-dopa – which connectivity measures would one use?). Third, the connectome for any given patient will be dependent on their individual disease progression; it would not allow us to assess disease propagation at a population-level as we are doing here. In this case, using a healthy brain connectome creates a basis to allow us to conduct between-subject comparisons.

6) I have a less substantive concern around the interpretation of the CSF beta-amyloid correlation with cortical thinning; what is the basis for this? One possibility (which the authors acknowledge) is that this was driven by coincident Alzheimer pathology. It is not clear how many patients in the Parkinson's cohort might have met CSF criteria for underlying Alzheimer pathology but at the very least this possibility tends to confound conclusions about the effect of Parkinson's disease per se.

The reviewer raises an interesting point regarding concurrent amyloid pathology which may confound our results. Using a previously suggested CSF amyloid-beta cut-off from McMillan & Wolk (2016), we found that 5 of 105 PD subjects met the amyloid-positive criterion. This sample size is too small for further investigation to compare potential differences but suggests that the large majority of our data is not confounded by underlying Alzheimer pathology. The larger and perhaps more interesting question on the interplay between AD and PD pathology is whether the two proteins interact e.g. does misfolded synuclein trigger an amyloidopathy which then leads to AD-like dementia in PD? It is the case that low CSF beta-amyloid or high brain amyloid by PET imaging predicts future dementia in PD (as cited in the discussion: Irwin 2012, Alves 2014 and Gomperts 2016). McMillan and Wolk (2016) studying the entire PPMI sample found that low CSF amyloid beta 42 subjects had lower cognitive function and greater gray matter atrophy in a pattern that included our two frontal clusters. An interesting interpretation then is that AD features (CSF, PET, brain atrophy) in PD are not a “confound” but possibly a part of the disease spectrum. A full investigation of the pathophysiology of amyloid beta in PD, and whether or not the abnormalities reported constitute actual AD in addition to PD, are beyond our scope here. The following text has been added to address this comment in the Discussion: *“While it can be argued that the findings observed in the present study may be driven by coincidental occurrence of Alzheimer's Disease, we found that only 5 of 105 PD subjects met a proposed CSF amyloid-positive criterion(McMillan & Wolk, 2016).”*

Reviewer #2 (Remarks to the Author):

This paper's major claim is that the atrophy pattern observed within the first year of PD diagnosis is correlated with their novel disease exposure metric (a sum of the atrophy in connected regions, weighted by the functional or structural connectivity strengths). They also perform path modeling to tease apart the relationship of CSF biomarkers, atrophy and measures of cognitive decline. I believe this is the first study of the effect of connectivity on the spatial pattern of disease in PD. I applaud the premise of the paper and believe that the results, if indeed true, are interesting and bring PD into the same fold as other dementias that have been shown to be prion-like. The analysis is generally well done, except for the apparent lack of multiple comparisons corrections.

1) Controlling for multiple comparisons is especially important here, as many of the reported significant p-values are close to (or exceed) the threshold of 0.05 (some tests used a threshold of 0.1 for unclear reasons). My concern is that the results may not survive the level of significance if MC-corrected. In particular, there are many correlations being performed in the analysis presented on page 10 (correlation of MOCA/CSF/regional cortical thinning measures). In addition, the uncorrected p-values for the beta coefficients in the path model are either close to the 0.05 level of significance (or above it).

We agree with the reviewer that after multiple comparison, some of the relationships between cortical atrophy, CSF levels, and cognitive decline (as assessed by MoCA) do not survive significance test. However, these results are not the main objective of the current paper. The key finding, that cortical atrophy over time is best predicted by disease spread via the connectome, safely survives multiple comparison testing (p values between 0.0004 and < 0.0001 by permutation testing). The cortical thinning at follow-up in PD vs Controls also survives multiple comparison testing over the entire cortical surface (Figure 2). These are the major results of this paper.

Nonetheless, the secondary analysis, which addresses the question of whether cortical thinning is a precursor of cognitive impairment, yields some findings of borderline statistical significance, although the overall model testing for an effect of cortical thinning in the four significant clusters is statistically significant (see below). To emphasize this fact, we have added the following text:

“Within PD, greater reduction in cortical thickness of the left frontal cluster over the one-year period was associated with worsening cognition (as assessed by the MoCA), although the p value of 0.03 would not survive correction for multiple comparisons. This mirrors results from other studies that have similarly

found trending results (Pereira et al., 2014; Caspell-Garcia et al., 2017). While dementia is rare in early PD, especially as our cohort was selected to exclude dementia at baseline, we still see greater cortical thinning of frontal clusters in the PD group one-year post-diagnosis...Though our results are not conclusive, they suggest that $A\beta_{42}$ may make the PD brain more susceptible to cortical thinning and in turn, more prone to cognitive decline."

We also tempered the paragraph of the discussion that discusses potential overlap with AD pathology, and start with: *"In this dataset only cortical thinning of the left frontal cluster related to cognitive decline. Furthermore, thinning in this frontal cluster was weakly related to reduced $A\beta_{42}$ at baseline (Figure 4)."*

We have also removed the text: *"Indeed, dementia in PD has a point prevalence of roughly 30% and incidence rates four to six times higher than controls (Aarsland & Kurz, 2010). Our results suggest that even in a dementia-free cohort of de novo PD patients, cortical atrophy and cognitive impairment may begin within one to two years of diagnosis."*

Furthermore, after reviewing our analysis and in consideration of the reviewers' comments, we concluded that the path model may not be an appropriate analysis for our dataset to address the relation between CSF markers, cortical thinning, and cognitive decline. Instead, we have conducted two separate analyses.

First, to examine whether $A\beta_{42}$ may relate to any of the four identified cortical clusters, we performed partial correlations using age and gender as covariates. We have replaced our previous analysis in the results section with the following text: *"To test whether CSF measures may contribute to the cortical thinning within each of the clusters, we conducted partial correlations (controlling for age and gender). We found changes in the left frontal ($r=-0.16$, $p=0.05$) and left occipital cluster ($r=-0.18$, $p=0.037$) both correlated with CSF $A\beta_{42}$. Changes of the left occipital cluster ($r=0.17$, $p=0.043$) also correlated with CSF α -syn."*

Second, to examine whether brain-related change relates to cognitive decline, we performed a linear regression model again including age and gender as covariates. As before, we replaced the previous analysis with the following text: *"We then performed a linear regression model to examine whether cortical thinning was related to cognitive decline (as assessed by MoCA) and found the overall model was significant ($r^2=0.12$, $p=0.028$). Further post-hoc analysis revealed that only the cortical thinning of the left frontal cluster ($r=-0.19$, $p=0.03$) significantly related to MoCA changes. Note that these analyses fail to survive correction for multiple comparisons of the significant 4 clusters."*

In sum, the key findings that test the network spread model are statistically valid, while the secondary analysis on the relation to cognitive decline is presented with caveats. We modified the Abstract by

replacing the final path model section with: *“The atrophy pattern in the ventral frontal lobes resembled one described in certain cases of Alzheimer’s disease. Moreover, a multiple linear regression model suggested that cortical thinning was associated with impaired cognitive function at follow-up. Our findings suggest that disease propagation to the cortex in PD follows neural connectivity, and that disease spread to the cortex may herald the onset of cognitive impairment.”*

2) I think it would also be interesting to investigate the relationship between disease burden and atrophy as measured using both structural and functional connectivity, as it may be the case that PD spread depends on both networks. It would be interesting to see a multivariate model incorporating the disease exposure based on both functional connectivity and structural connectivity predicting atrophy. That way, we could compare the relative contribution of both networks to disease spread. The authors may also consider adding demographics to such a multivariate model to see if age/gender has any impact on the correlations observed in Figure 5.

We agree with the reviewer that it will be of interest to compare the relative contribution of functional and structural connectivity in predicting atrophy. As the reviewer suggested, we conducted a multivariate model for each hemisphere and found that both functional and structural connectivity significantly predicted cortical atrophy. We have added the following text to the results section: *“We also tested the relative contribution of functional and structural networks to disease spread using a multivariate model for each hemisphere. Within the left hemisphere, the analysis shows that both functional connectivity ($\beta = .18$, $t(220)=-3.88$, $p<0.001$) and structural connectivity ($\beta = .04$, $t(220)=-2.27$, $p=0.02$) significantly relate to cortical atrophy. Within the right hemisphere, again functional connectivity ($\beta = .12$, $t(220)=-2.56$, $p=0.01$) and structural connectivity ($\beta = .08$, $t(220)=-2.77$, $p<0.01$) significantly relate to cortical atrophy. These results suggest that both types of connectivity play a role in the disease spread model, however, functional connectivity appears to be more significant.”* We did not run a multivariate model at a whole brain level due to the limitations of structural connectivity to model interhemispheric connections. We detail this shortcoming in the reply to question 3 immediately below.

Regarding the impact of age and gender, they are controlled for when obtaining both cortical thickness and connectivity measures; we have added the following text to clarify this, *“Age at baseline and gender were used as confounding covariates for all analyses including for partial correlations, linear regression model, and for obtaining both cortical thickness.”*

3) It is unclear what exactly is being done in the correlation between disease exposure and atrophy – what is the difference in the whole brain correlations versus the right and left hemisphere only? Are the disease burden values being calculated using only regions in the left hemisphere (i.e. if there is a connected region in the right hemisphere it is ignored)? If the latter is the case, why would the problems with tractography not picking up inter-hemispheric connections result in a decrease of correlation in the whole brain model? In the worst case scenario, we wouldn't have any connections between right and left hemispheres and would be left with the values that are in the separate hemisphere analysis. Please provide more detail.

As the reviewer points out, we first use models with regions within one hemisphere when calculating the disease exposure – cortical thinning comes from one hemisphere at a time. We then use a whole brain model. This doubles the number of regions in the model. However, due to the nature of diffusion weighted tractography (DTI) and its poor ability to correctly trace inter-hemispheric connection across the corpus callosum, it can result in bias towards incorrect inter-hemispheric connectivity estimates. It isn't that DTI fails to identify inter-hemispheric projections (false negatives), but that it probably identifies incorrect connections (false positives). We agree if DTI detected no inter-hemispheric connections at all the whole brain model would end up being an average of the two hemisphere-only models. However, the known presence of false positives across the corpus callosum likely explains the lack of correlation in the whole brain DTI model.

This insensitivity to inter-hemispheric connectivity in DTI has been previously detailed and we have added citations to previous work by Thomas *et al.* (2014) and Neher *et al.* (2015) to the paper. In contrast, functional connectivity is better able to capture inter-hemispheric connectivity. We believe this is probably the reason we see a significant correlation between regional cortical thinning and disease exposure across the whole-brain (i.e., inter-hemispherically) when using functional connectivity and not when using structural connectivity.

We have added the following text to the discussion addressing this issue “*While functional connectivity models fit the data at a whole-brain level, we only observed a relationship between connectivity and thinning in individual hemispheres when using a DW-MRI connectome. This is perhaps due to inherent limitations in DW-MRI to capture interhemispheric connections (Jones et al., 2013; Thomas et al., 2014)*”

4) In Figure 5 c and d, it appears that the disease exposure value is not Gaussian (nor should it be, since it depends on structural connectivity), so Spearman's correlation is probably more

appropriate here. It also appears that the correlation is being driven by a handful of regions that are highly connected to the subcortical areas – which are these?

We thank the reviewer for noting this detail and agree our disease exposure value is not Gaussian. Spearman's correlation should have been conducted instead of Pearson's and we have replaced the statistics and text where appropriate in the manuscript. This did not change any of our findings, with regards to either directionality or significance. Below is an excerpt of the two paragraphs where we have updated the correlation statistics:

“The analysis was performed for the whole brain, and for each hemisphere separately using Spearman's correlation to examine the relation between disease progression and exposure...First, a functional connectome was generated using resting state fMRI to define Connij. Significant correlation was observed between regional cortical thinning and disease exposure ($r=0.19$, $p<0.0001$). Significance was confirmed by permutation testing where we kept the connectivity structure (i.e. subcortical-cortical connections) intact and permuted the cortical thinning values ($n=10,000$) and the 95% confidence interval was measured using bootstrapping ($n=10,000$) ($r \in [0.11, 0.28]$, $p_{perm}<0.0001$). Moreover, cortical thickness of both the left ($r=0.22$, $p=0.0005$, $r \in [0.09, 0.34]$, $p_{perm}=0.0004$) and right hemisphere ($r=0.27$, $p<0.0001$, $r \in [0.16, 0.38]$, $p_{perm}<0.0001$) significantly correlated with the disease exposure from bilateral affected regions.

Second, the same hypothesis was tested using structural connectivity as measured by diffusion-weighted MRI (DW-MRI). Though the correlation was not significant at the whole-brain level ($r= -0.03$, $p=0.239$, $r \in [-0.12, 0.06]$, $p_{perm}=0.762$), change in cortical thickness and disease exposure within the individual hemispheres were significantly correlated. Similar results were observed within the left hemisphere ($r=0.14$, $p=0.016$, $r \in [0.01, 0.27]$, $p_{perm}=0.016$) and the right hemisphere ($r=0.11$, $p=0.044$, $r \in [-0.02, 0.25]$, $p_{perm}=0.041$). Failure to capture the relationship at the whole brain level may be due to inherent limitations in DW-MRI to detect interhemispheric connectivity.”

To test whether there were some regions that may drive the correlation between disease exposure and cortical thickness, we looked at potential outliers defined by QQ-plots (Figure Sup1). With this, we did not find any particular set of regions driving the effect in the functional or structural connectivity analysis. Nonetheless, to check for effects of potential outliers, we removed regions that fell in the tails of the QQ-plots and found that the relationship between disease exposure and cortical thickness remained significant for both functional connectivity: (left, $r=0.22$, $p=0.0005$; right, $r=0.28$, $p<0.0001$) and structural connectivity: (left, $r=0.19$, $p=0.003$; right, $r=0.17$, $p=0.005$).

In addressing which areas are may be more connected to the subcortical areas, we identified the regions most connected to the subcortical areas based on functional and structural connectivity independently. Interestingly, using structural connectivity, these identified regions coincided with the frontal clusters previous identified in the cortical thinning analysis between PD and controls. For functional connectivity, the regions are more diverse but largely restricted to temporal and frontal regions. These regions are as follows for (i) structural connectivity: rostral middle frontal, frontal pole, superior frontal, and medial orbitofrontal; and (ii) functional connectivity: lateral orbitofrontal, inferior temporal, entorhinal, and fusiform.

Figure Sup1. *QQ-plots for distribution of data points for functional and structural connectivity within the left and right hemisphere.*

5) From the color scale on Figure 5, it appears that cortical atrophy would be correlated with just subcortical connectivity, not connectivity modulated by atrophy? Does the atrophy of the subcortical region increase the correlation?

When removing subcortical atrophy and only considering connectivity, we observed there were some changes in correlation strength between connectivity to the disease reservoir and cortical atrophy (Table Sup2; please refer to Table Sup1 addressing Reviewer 1’s comment for the original analysis not restricted to connectivity to subcortical areas), though this change in correlation value does not appear to be systematic. Subcortical atrophy values in different regions are relatively homogenous and therefore, ubiquitously affect cortical thinning via the brain’s intrinsic connectome. As such, there is not enough variance between the different subcortical structures to systematically affect our correlation results. In other words, no one region is driving the estimated disease exposure and therefore, connectivity to subcortical areas in general is a good representative of the disease exposure. We thank the reviewer for highlighting this point; to not mislead the readers, we have removed the colour scale.

Table Sup2. Summary table of Spearman correlations between subcortical connectivity and progression

	r-value	CI 95% lower	CI 95% upper	p-value	p-perm
Functional Connectivity to Subcortical Areas					
bilateral	0.24	0.16	0.32	0.0000	0.0000
right hemisphere	0.24	0.13	0.35	0.0001	0.0001
left hemisphere	0.24	0.12	0.36	0.0002	0.0001
Structural Connectivity to Subcortical Areas					
bilateral	0.07	-0.03	0.16	0.0754	0.0784
right hemisphere	0.11	-0.03	0.25	0.0511	0.0475
left hemisphere	0.09	-0.05	0.23	0.0842	0.0835

6) Finally, the title of the paper is misleading. From what I can tell, no associations were found between network-based metrics and cognitive decline. Associations were found between cortical thinning and MOCA scores, but this had nothing to do with network connectivity. Actually, this would be interesting if you also did a path model that not only looked at change in cortical thickness in the four clusters, but change in cortical thickness and that region's overall connectivity to the subcortical structures.

Indeed, we agree with the reviewer that it would be interesting to investigate how alterations in connectivity may modulate/affect our path model analysis. A problem with the proposed path model is that the paths between cortical thickness and MoCA are subject-level, while the path between changes in cortical thickness and connectivity are at the group-level. One strength of our study is that we use a healthy connectome that allows us to derive measures in a group of individuals with PD. As mentioned in a reply to Reviewer 1's comment, the connectome for any given patient is dependent on their individual disease stage and medication use and therefore cannot easily be incorporated in our model. An alternative would be to compare changes in the structural connectome in PD patients (for which the PPMI has data) from one derived from healthy subjects. However, this is beyond the scope of the current paper and has been previously addressed with the PPMI dataset by another group (Zhang *et al.*, 2015).

We also agree with the comment on our title. We have changed the title to "Network Connectivity Determines Cortical Thinning in Early Parkinson's Disease Progression".

7) There is a relative lack of description of the structural connectivity. I realize that the data being used has been published, but it may make sense to provide the bare minimum description of SC network construction (is it number of streamlines? What was the number of directions/resolution of the dMRI? What tractography algorithm was used?)

The approach is based on a probabilistic graph-based tractography algorithm described in Iturria-Medina *et al.* (2007). We added further text to describe how the structural connectome was constructed:

"The structural connectivity map was generated using the Illinois Institute of Technology Human Brain Atlas v.3, (freely available at <http://www.iit.edu/~mri/Home.html>). The atlas was constructed using data from 72 neurologically healthy subjects (20-40 yr old, 28.8 ± 5.5 yr old, 30 males). All subjects were scanned using a 3T General Electric MRI scanner (GE, Waukesha, WI) with the following parameters (TR = 5800 ms, TE = 94 ms, voxel size = 3 mm³, 256×256 final image matrix, FoV = 24 × 24 cm, b = 900 s/mm², 45 slices for 12 diffusion directions uniformly distributed in 3D space, scan-time: 21 min and 57 s) (Varentsova *et al.*, 2014). Imposing a maximum of 500-mm trace length and a curvature threshold

of ± 90 , the anatomical connection probability (ACP) was calculated between all pairs of regions using a probabilistic graph-based tractography algorithm (Iturria-Medina *et al.*, 2007).”

Reviewer #3 (Remarks to the Author):

This manuscript attempts to examine the longitudinal cortical thinning patterns in PD and test its relationship with cognitive functions and network spread hypothesis. Although it is highly motivated, the work has several issues in methods, literature, and interpretations, which reduce the confidence of its conclusion.

1. Abstract: the authors wrote that ‘The atrophy pattern in the frontal lobes resembled that described in Alzheimer’s Disease’. I am concerned about this statement. Frontal lobes are not the typical regions atrophied in AD. In addition, it is written in the last sentence ‘cognitive impairments occur earlier than previously thought in PD’. It is unclear which evidence in the manuscript support this claim.

We agree that both claims need to be better stated. First, the frontal lobes are not typically thought to be an early atrophy target in AD. Nonetheless, the observed pattern of frontal atrophy in PD described here is similar to a previously reported subtype in Alzheimer’s Disease (Hwang *et al.*, 2016). These authors performed hierarchical clustering on cortical thickness data from ADNI, and found an inferior frontal pattern in 19.5% of patients. For reference, we have copied a figure from their paper below (Figure Sup2). The inferior frontal pattern was also described by Seeley *et al.* (2009) in patients with the behavioral variant frontotemporal dementia phenotype (a rarer condition than AD). Finally, we note that in the Braak model of AD, the initial cortical site of amyloid- β involvement is the ventral frontal cortex (cited in the manuscript). In sum, involvement of the inferior frontal lobes is described in amyloidopathies and in PD and may account for a specific constellation of behavioral and cognitive deficits.

Please see Figure 1 panel C from Hwang *et al.* 2016

2. Some key work in network spread hypothesis in the neuroimaging fields should be cited (e.g. Raj Neuron 2012, Zhou Neuron 2012, Iturria-Medina et al, PLOS Computational Biology, 2014).

We previously only cited Raj and Iturria-Medina’s work in the discussion but have now added the citation for Zhou *et al.* (2012) to the sentence: “Mathematical models of this dynamic spread through brain networks, accounting for factors such as protein accumulation and clearance, have been proposed in recent years for Alzheimer’s Disease”

3. I may not fully agree with the comment that ‘cortical thickness analysis may be better able to delineate cortical morphometrical patterns and more sensitive to age or disease-related regional grey matter changes than VBM’. Each method could pick up different patterns of change in disease, depending on the sample, disease etc.

We agree with the reviewer that VBM may be as good as cortical thickness in examining volumetric aspects of brain. Both methods have strengths and limitations. Here, the goal of the study is to identify cortical morphometrical alterations for which cortical thickness measurements may be more sensitive. There is a growing body of research suggesting a structural analysis technique that uses MRI to assess local cortical folding and geometrical properties can help elucidate neuronal degeneration in PD. While previous studies using VBM techniques to assess grey matter (GM) density and atrophy based on delineation of GM and normalization have begun to highlight a pattern of cortical changes in PD (Reetz *et al.*, 2010; Fioravanti *et al.*, 2015), there have been discrepancies in findings (Camicioli *et al.*, 2009; Menke *et al.*, 2014). Such mixed results are in-part due to heterogeneity in disease progression but are also due to methodological issues (e.g., non-specific spatial smoothing (Davatzikos, 2004)). Cortical thickness analysis may be better able to delineate cortico-morphometrical patterns and may be more sensitive to age-related regional GM changes and neurodegeneration in PD (Hutton *et al.*, 2009; Pereira *et al.*, 2012). Cortical thickness therefore holds potential as an effective biomarker that can robustly track and predict PD progression patterns. We have briefly detailed this in the introduction but the discussion of which method is superior is beyond the scope of this paper.

4. In Figure 1, what is the voxel and cluster level threshold for regional cortical thinning comparison between PD and HC? Did the authors check the opposite direction, i.e., (HC1-HC2)-(PD1-PD2) at the same threshold? What are the covariates included in the statistical analysis?

We detail the statistical analysis for voxel and cluster level cortical thickness analysis in the methods section: “Difference between PD and controls [(PD1-PD2)-(HC1-HC2)] were analyzed statistically based on random field theory with a threshold of $p < .05$ corrected for multiple comparisons over the entire cortical surface”.

With regards to covariates, we have also detailed in the method that “Acquisition site was used as a covariate in all analyses.” and that “Age at baseline and gender were used as confounding covariates for all analyses”. However, in order to address this comment and a previous comment from Reviewer 2, we have added the text: “Age at baseline and gender were used as confounding covariates for all analyses including for partial correlations, linear regression model, and for obtaining both cortical thickness.”

When checking for potential changes in the opposite direction, that is testing whether areas might be thinning in healthy controls more than PD patients over the one-year, we observed no significant differences. This is in the text now.

5. It is unclear to me that whether Fig. 3 is derived from the Yeo's parcellation directly or some clusters showing significant group difference. There is no mention of the rationale of such analysis and its relationship to results presented in Figure 1 & 2.

Our analysis is from Yeo et al's parcellation directly, that is, we overlaid Yeo's parcellation over the 463 regions that were used for connectivity analysis without any consideration of significant clusters. After assigning each of our regions to one of Yeo's parcels, we calculated the effect size of the group difference and results are depicted in Figure 3.

In order to better describe which brain networks may be implicated in the disease progression of PD, we chose Yeo's parcellation as a descriptor due to its eminence in the literature. These networks are thought to be building blocks of higher cognitive function hence interesting. This gave us a clearer view of how regional cortical thickness changes may be affected from the perspective of intrinsic brain networks. We believe this provides interesting complementary information in addition to a statistical map. Furthermore, this allows us to compare with previous literature on Alzheimer's Disease where there is strong evidence for alterations in the default-mode and frontoparietal networks. We have added the following text to the results section: *"In order to assess the likely functional consequences of cortical thinning in PD, we assigned each cortical parcel to one of the seven intrinsic brain networks derived from resting state fMRI by Yeo et al. (2011). Cortical regions belonging to the limbic, frontoparietal, and ventral attention networks on average demonstrated the greatest cortical thinning, as well as anterior parts of the default mode network (Figure 3)."*

6. My major concern is regarding the analysis between CSF, cortical thickness, and MoCA scores. Some p-values are not even significant or only marginal ($p < 0.1$). Because CSF does not relate to MoCA in the first place, it is hard to interpret the path model of CSF-cortical thinking-MoCA here and thus the conclusion of synergistic effect with Ab42 is questionable.

We thank the reviewer for this important comment. After reviewing our path model, we agree with the reviewer's concern that the path model may not be an appropriate analysis for our dataset to address the questions at hand. Instead, we have now conducted two separate analyses.

First, to examine whether $A\beta_{42}$ may relate to any of the four identified cortical clusters, we performed partial correlations using age and gender as covariates. We have replaced our previous analysis in the results section with the following text: *“To test whether CSF measures may contribute to the cortical thinning within each of the clusters, we conducted partial correlations (controlling for age and gender). We found changes in the left frontal ($r=-0.16$, $p=0.05$) and left occipital cluster ($r=-0.18$, $p=0.037$) both correlated with CSF $A\beta_{42}$. Changes of the left occipital cluster ($r=0.17$, $p=0.043$) also correlated with CSF α -syn.”*

Second, to examine whether cortical thinning relates to cognitive decline, we performed a linear regression model again including age and gender as covariates. As before, we replaced the previous analysis with the following text: *“We then performed a linear regression model to examine whether cortical thinning was related to cognitive decline (as assessed by MoCA) and found the overall model was significant ($r^2=0.12$, $p=0.028$). Further post-hoc analysis revealed that only the cortical thinning of the left frontal cluster ($r=-0.19$, $p=0.03$) significantly related to MoCA changes. Note that these analyses fail to survive correction for multiple comparisons of the 4 significant clusters.”*

We have correspondingly changed the methods section to detail our new analyses: *“Observed differences in cortical thickness were then correlated to other potential disease-related measures (i.e., UPDRS-III, MoCA, CSF α -syn, and tau and $A\beta_{42}$ markers) using Pearson partial correlations. Within PD patients, the difference in cognitive performance over the one-year interval was examined using a linear regression model with MoCA as the dependent variable and the thinning of the four cortical clusters as independent variables. To test whether cortical thickness of a specific cluster was linked to cognitive performance, post-hoc analyses using partial correlations were conducted. Age at baseline and sex were used as confounding covariates for all analyses including for partial correlations, linear regression model, and for obtaining cortical thickness.”*

Furthermore, we have changed our discussion to reflect these new analyses and have removed all text that may implicate directionality. We have also added the following text to our discussion to emphasize the limitations of our results: *“Within PD, greater reduction in cortical thickness of the left frontal cluster over the one-year period was associated with worsening cognition (as assessed by the MoCA), although the p value of 0.03 would not survive correction for multiple comparisons. This mirrors results from other studies that have similarly found trending results (Pereira et al., 2014; Caspell-Garcia et al., 2017). While dementia is rare in early PD, especially as our cohort was selected to exclude dementia at baseline, we still see greater cortical thinning of frontal clusters in the PD group one-year post-diagnosis... Though our results are not conclusive, they suggest that $A\beta_{42}$ may make the PD brain more susceptible to cortical thinning and in turn, more prone to cognitive decline.”*

Finally, we amended the title and changed the final part of the abstract to: *“The atrophy pattern in the ventral frontal lobes resembled one described in certain cases of Alzheimer’s disease. Moreover, a multiple linear regression model suggested that cortical thinning was associated with impaired cognitive function at follow-up. Our findings suggest that disease propagation to the cortex in PD follows neural connectivity, and that disease spread to the cortex may herald the onset of cognitive impairment*

7. Lastly, the network spread hypothesis is only weakly supported here. The r-values between disease exposure and atrophy is only 0.15-0.25. As the authors examined cortical thickness, there is no way to evaluate subcortical atrophy (i.e., substantia nigra) changes over time. It is actually important to characterise the longitudinal changes of this so-called disease onset region or reservoir first and then compare it with the rest of the region. VBM approach may be a good alternative to validate the findings here.

It is likely that multiple factors contribute to the progression trajectory in PD. Our cortical atrophy measure is an indirect way of trying to capture disease progression. The hypothesis is that misfolded synuclein spreads via the connectome, and triggers the formation of more misfolded synuclein, which eventually leads to cell loss. However we cannot measure synuclein accumulation, but only its consequence (cortical thinning). Further, cellular factors (e.g. autophagy, synuclein synthesis and removal) may determine the local vulnerability to neurotoxicity. These local factors would have the effect of mitigating the strength of the correlation between connectivity and thinning. Nonetheless, we think that the fact that functional connectivity explains cortical thinning better than chance, and better than physical proximity, supports the network spread hypothesis in spite of the small effect sizes. We add the following to the limitations section:

“Moreover, we cannot measure α -syn accumulation, but only its consequence (cortical thinning), and the two may have a complex and non-linear relationship.”

As the reviewer noted, we cannot evaluate subcortical atrophy as we are looking in surface space. We plan to look at the progression of subcortical atrophy in future work, but this particular study tests spread to cortex and its relation to cognitive impairment.

8. Also, there is no figure on the disease reservoir. Structural connectivity and functional connectivity preprocessing and connectivity definitions should be provided in details to allow better evaluation of the results. Given the image resolution, the structural and functional connectivity to such small subcortical region (substantia nigra) might be challenging and not sufficiently reliable.

We agree that adding a figure of the disease reservoir would assist our readers. As such, we have added both a schematic and an MRI image of the PD-ICA atrophy map used to define our disease reservoir to Figure 5, which we have also attached below (Figure Sup3).

With respect to the reviewer's concern about connectivity to small subcortical regions like the substantia nigra, our disease reservoir is not limited to this one region. The disease reservoir consisted of all of the 15 subcortical regions from the Desikan-Killiany Atlas. In each region, the disease quantity was derived from the atrophy map from Zeighami et al. 2015. We now state more explicitly the regions that define the reservoir:

“based on a parcellation that is a subdivision of the common FreeSurfer implementation of the Desikan-Killiany Atlas, grey matter was segmented into 463 regions of interest of approximately similar size. Of these parcels, 448 were cortical and 15 were subcortical (*thalamus, caudate, putamen, pallidum, accumbens, hippocampus, amygdala in each hemisphere and brainstem*)²⁸. Each surface vertex from the cortical thickness analysis was then mapped to the corresponding nearest whole-brain parcel from the Atlas. The “disease reservoir” was defined as the 15 subcortical parcels from the Atlas, and the Atrophy value at each parcel was obtained from the DBM map.”

Figure Sup3. (a) Schematic of the propagation model. The model used a disease reservoir (colored regions) consisting of 15 subcortical regions of interest and connectivity from each region to the cortical surface. (b) PD-specific atrophy map used to define disease “content” in each region of interest of the disease reservoir, derived from deformation based morphometry analysis of patients at baseline (Zeighami et al. 2015).

References:

- Aarsland, D. & Kurz, M.W. (2010) The epidemiology of dementia associated with Parkinson disease. *Journal of the Neurological Sciences*, **289**, 18-22.
- Akram, H., Wu, C., Hyam, J., Foltynie, T., Limousin, P., De Vita, E., Yousry, T., Jahanshahi, M., Hariz, M., Behrens, T., Ashburner, J. & Zrinzo, L. (2017) l-Dopa responsiveness is associated with distinctive connectivity patterns in advanced Parkinson's disease. *Mov Disord*, **32**, 874-883.
- Camicioli, R., Gee, M., Bouchard, T.P., Fisher, N.J., Hanstock, C.C., Emery, D.J. & Martin, W.R.W. (2009) Voxel-based morphometry reveals extra-nigral atrophy patterns associated with dopamine refractory cognitive and motor impairment in parkinsonism. *Parkinsonism & Related Disorders*, **15**, 187-195.
- Caspell-Garcia, C., Simuni, T., Tosun-Turgut, D., Wu, I.W., Zhang, Y., Nalls, M., Singleton, A., Shaw, L.A., Kang, J.-H., Trojanowski, J.Q., Siderowf, A., Coffey, C., Lasch, S., Aarsland, D., Burn, D., Chahine, L.M., Espay, A.J., Foster, E.D., Hawkins, K.A., Litvan, I., Richard, I., Weintraub, D. & the Parkinson's Progression Markers, I. (2017) Multiple modality biomarker prediction of cognitive impairment in prospectively followed de novo Parkinson disease. *PLOS ONE*, **12**, e0175674.
- Davatzikos, C. (2004) Why voxel-based morphometric analysis should be used with great caution when characterizing group differences. *NeuroImage*, **23**, 17-20.
- Engelender, S. & Isacson, O. (2017) The Threshold Theory for Parkinson's Disease. *Trends Neurosci*, **40**, 4-14.
- Ercsey-Ravasz, M., Markov, Nikola T., Lamy, C., Van Essen, David C., Knoblauch, K., Toroczkai, Z. & Kennedy, H. (2013) A Predictive Network Model of Cerebral Cortical Connectivity Based on a Distance Rule. *Neuron*, **80**, 184-197.
- Fioravanti, V., Benuzzi, F., Codeluppi, L., Contardi, S., Cavallieri, F., Nichelli, P. & Valzania, F. (2015) MRI Correlates of Parkinson's Disease Progression: A Voxel Based Morphometry Study. *Parkinson's Disease*, **2015**, 8.
- Gong, G., He, Y., Chen, Z.J. & Evans, A.C. (2012) Convergence and divergence of thickness correlations with diffusion connections across the human cerebral cortex. *Neuroimage*, **59**, 1239-1248.
- Hutton, C., Draganski, B., Ashburner, J. & Weiskopf, N. (2009) A comparison between voxel-based cortical thickness and voxel-based morphometry in normal aging. *Neuroimage*, **48**, 371-380.
- Hwang, J., Kim, C.M., Jeon, S., Lee, J.M., Hong, Y.J., Roh, J.H., Lee, J.-H. & Na, D.L. (2016) Prediction of Alzheimer's disease pathophysiology based on cortical thickness patterns. *Alzheimer's & Dementia: Diagnosis, Assessment & Disease Monitoring*, **2**, 58-67.
- Iturria-Medina, Y., Canales-Rodríguez, E.J., Melie-García, L., Valdés-Hernández, P.A., Martínez-Montes, E., Alemán-Gómez, Y. & Sánchez-Bornot, J.M. (2007) Characterizing brain anatomical connections using diffusion weighted MRI and graph theory. *NeuroImage*, **36**, 645-660.
- Jbabdi, S., Sotiropoulos, S.N., Haber, S.N., Van Essen, D.C. & Behrens, T.E. (2015) Measuring macroscopic brain connections in vivo. *Nat Neurosci*, **18**, 1546-1555.

- Jones, D.K., Knosche, T.R. & Turner, R. (2013) White matter integrity, fiber count, and other fallacies: the do's and don'ts of diffusion MRI. *Neuroimage*, **73**, 239-254.
- McMillan, C.T. & Wolk, D.A. (2016) Presence of cerebral amyloid modulates phenotype and pattern of neurodegeneration in early Parkinson's disease. *Journal of Neurology, Neurosurgery & Psychiatry*, **87**, 1112-1122.
- Menke, R.A., Szewczyk-Krolikowski, K., Jbabdi, S., Jenkinson, M., Talbot, K., Mackay, C.E. & Hu, M. (2014) Comprehensive morphometry of subcortical grey matter structures in early-stage Parkinson's disease. *Human brain mapping*, **35**, 1681-1690.
- Neher, P.F., Descoteaux, M., Houde, J.-C., Stieltjes, B. & Maier-Hein, K.H. (2015) Strengths and weaknesses of state of the art fiber tractography pipelines – A comprehensive in-vivo and phantom evaluation study using Tractometer. *Medical Image Analysis*, **26**, 287-305.
- Pereira, J.B., Ibarretxe-Bilbao, N., Marti, M.J., Compta, Y., Junque, C., Bargallo, N. & Tolosa, E. (2012) Assessment of cortical degeneration in patients with Parkinson's disease by voxel-based morphometry, cortical folding, and cortical thickness. *Hum Brain Mapp*, **33**, 2521-2534.
- Pereira, J.B., Svenningsson, P., Weintraub, D., Bronnick, K., Lebedev, A., Westman, E. & Aarsland, D. (2014) Initial cognitive decline is associated with cortical thinning in early Parkinson disease. *Neurology*, **82**, 2017-2025.
- Reetz, K., Tadic, V., Kasten, M., Bruggemann, N., Schmidt, A., Hagenah, J., Pramstaller, P.P., Ramirez, A., Behrens, M.I., Siebner, H.R., Klein, C. & Binkofski, F. (2010) Structural imaging in the presymptomatic stage of genetically determined parkinsonism. *Neurobiol Dis*, **39**, 402-408.
- Richiardi, J., Altmann, A., Milazzo, A.C., Chang, C., Chakravarty, M.M., Banaschewski, T., Barker, G.J., Bokde, A.L., Bromberg, U., Buchel, C., Conrod, P., Fauth-Buhler, M., Flor, H., Frouin, V., Gallinat, J., Garavan, H., Gowland, P., Heinz, A., Lemaitre, H., Mann, K.F., Martinot, J.L., Nees, F., Paus, T., Pausova, Z., Rietschel, M., Robbins, T.W., Smolka, M.N., Spanagel, R., Strohle, A., Schumann, G., Hawrylycz, M., Poline, J.B. & Greicius, M.D. (2015) Correlated gene expression supports synchronous activity in brain networks. *Science*, **348**, 1241-1244.
- Seeley, W.W., Crawford, R.K., Zhou, J., Miller, B.L. & Greicius, M.D. (2009) Neurodegenerative diseases target large-scale human brain networks. *Neuron*, **62**, 42-52.
- Simioni, A.C., Dagher, A. & Fellows, L.K. (2016) Compensatory striatal–cerebellar connectivity in mild–moderate Parkinson's disease. *NeuroImage: Clinical*, **10**, 54-62.
- Surmeier, D.J., Obeso, J.A. & Halliday, G.M. (2017) Selective neuronal vulnerability in Parkinson disease. *Nat Rev Neurosci*, **18**, 101-113.
- Thomas, C., Ye, F.Q., Irfanoglu, M.O., Modi, P., Saleem, K.S., Leopold, D.A. & Pierpaoli, C. (2014) Anatomical accuracy of brain connections derived from diffusion MRI tractography is inherently limited. *Proc Natl Acad Sci U S A*, **111**, 16574-16579.

- Varentsova, A., Zhang, S. & Arfanakis, K. (2014) Development of a High Angular Resolution Diffusion Imaging Human Brain Template. *NeuroImage*, **91**, 177-186.
- Walsh, D.M. & Selkoe, D.J. (2016) A critical appraisal of the pathogenic protein spread hypothesis of neurodegeneration. *Nat Rev Neurosci*, **17**, 251-260.
- Yeo, B.T., Krienen, F.M., Sepulcre, J., Sabuncu, M.R., Lashkari, D., Hollinshead, M., Roffman, J.L., Smoller, J.W., Zöllei, L. & Polimeni, J.R. (2011) The organization of the human cerebral cortex estimated by intrinsic functional connectivity. *Journal of neurophysiology*, **106**, 1125-1165.
- Zhang, L., Wang, M., Sterling, N., Lee, E., Eslinger, P., Wagner, D., Du, G., Lewis, M., Truong, Y., Bowman, D. & Huang, X. (2015) Cortical Thinning and Cognitive Impairment in Parkinson's Disease Without Dementia *IEEE/ACM Transactions on Computational Biology and Bioinformatics*, **PP**, 1-1.
- Zhou, J., Gennatas, E.D., Kramer, J.H., Miller, B.L. & Seeley, W.W. (2012) Predicting regional neurodegeneration from the healthy brain functional connectome. *Neuron*, **73**, 1216-1227.

Reviewers' comments:

Reviewer #1 (Remarks to the Author):

I feel that, on balance, the authors have now adequately addressed my previous concerns.

Reviewer #2 (Remarks to the Author):

In general, the authors have addressed most of the concerns from the reviewers. My main remaining concern is two-fold: one is the weak to moderate strength of the correlations that remained significant after multiple comparisons (0.15-0.2). The other is in the linear regression modeling (see below). If the regression results were presented as an add-on post hoc analysis then it might be okay, but they are highlighted in the abstract even though the r-squared values were not very high and the results were not significant after multiple comparisons correction. This reduces my enthusiasm for the new analysis.

While the linear regression analysis is more appropriate in this case, there are very few details on the implementation and results which makes it difficult to assess the quality of the analysis. What were the inputs to the model (co-variates) and what atlas of cortical regions' thinning was included? What were the beta values for the regression coefficients? Is the r-squared value reported after removing non-significant regressors or was it done with all of the regressors included?

Reviewer #3 (Remarks to the Author):

The authors have made good effort addressing my questions. I have two more questions.

Because the authors used all 15 subcortical structures as the disease reservoir, the sum of connectivity x atrophy to all the 15 regions was used to relate to cortical thinning. Such approach might not be sensitive to the specific contribution of each disease onset region. It would be important and informative to compare the predictive value of these 15 regions. Or the authors could take the maximum value among the 15 regions and perform correlation analysis with cortical thinning.

Another concern is that striatum and brainstem might be the key region vulnerable in PD but amygdala and hippocampus/thalamus might not be. How to justify using all 15 regions as disease reservoir together? It might explain the relatively weak correlations ($r = 0.1 - 0.2$), which is concerned by all three reviewers here. If only striatum and midbrain are used, will the correlation results remain or even stronger? It is worth finding out.

Reviewer #1 (Remarks to the Author):

I feel that, on balance, the authors have now adequately addressed my previous concerns.

We thank the reviewer for their kind remarks.

Reviewer #2 (Remarks to the Author):

(1) In general, the authors have addressed most of the concerns from the reviewers. My main remaining concern is two-fold: one is the weak to moderate strength of the correlations that remained significant after multiple comparisons (0.15-0.2). The other is in the linear regression modeling (see below). If the regression results were presented as an add-on post hoc analysis then it might be okay, but they are highlighted in the abstract even though the r-squared values were not very high and the results were not significant after multiple comparisons correction. This reduces my enthusiasm for the new analysis.

To address the reviewer’s concern, we have further modified our Abstract to clarify that the regression results are indeed an add-on post hoc analysis and that effects were moderate. (However, it is important to note that the cortical thinning versus MoCA change model, controlling for covariates, was significant ($r^2=0.12, p=0.03$), when inputting all four significant clusters.)

“Because cortical atrophy might be expected to lead to cognitive impairment, we performed a multiple linear regression analysis between thinning in all four clusters and cognitive scores as a post-hoc analysis, but found only a weak relationship between cortical thinning and MoCA score at follow-up.”

We also highlight this in our Discussion in the following text:

“In our post-hoc analysis, greater reduction in cortical thickness of the left frontal cluster in PD over the one-year period was associated with worsening cognition (as assessed by the MoCA), although the p value of 0.03 would not survive correction for multiple comparisons. This mirrors results from other studies that have similarly found trending results^{1,2}. While dementia

is rare in early PD, especially as our cohort was selected to exclude dementia at baseline, we still see greater cortical thinning of frontal clusters in the PD group one-year post-diagnosis.”

(2) While the linear regression analysis is more appropriate in this case, there are very few details on the implementation and results which makes it difficult to assess the quality of the analysis. What were the inputs to the model (co-variates) and what atlas of cortical regions' thinning was included? What were the beta values for the regression coefficients? Is the r-squared value reported after removing non-significant regressors or was it done with all of the regressors included?

We agree that further details are needed for the linear regression model and have added these. Brain areas were restricted to cortical clusters determined by the cortical thickness analysis that compares PD and controls over one-year. Age and gender were used as co-variates. The r-squared value reported considers all the regressors (i.e., including age and gender). Out of the 12% covariance explained, 5% can be attributed to the left frontal cluster while 6% can be attributed to age. The linear regression model was implemented in MatLab using the fitLME function (MoCA ~ 1 + Cortical Cluster 1 + Cortical Cluster 2 + Cortical Cluster 3 + Cortical Cluster 4 + Age + Gender) and results are detailed in the Table Sup1 below.

	Estimate	SE	tStat	p-value
(intercept)	-0.19	0.17	-1.06	0.14
CT1 – L Occipital	0.05	0.12	0.44	0.32
CT2 – L Frontal	-0.23	0.13	-1.70	0.04
CT3 – R Frontal	0.02	0.13	0.18	0.42
CT4 – R S1/M1	0.07	0.11	0.64	0.25
Age	0.22	0.09	2.28	0.01
Gender	0.27	0.21	1.26	0.10

Table Sup1. Linear regression model. Significant findings are highlighted in bold.

We have added the following text to the manuscript:

Methods: *“In order to address the relationship between cognitive decline and cortical thinning over the one-year period, we performed a linear regression model. Change in MoCA was the dependent variable, while mean cortical thinning values from each of the four clusters distinguishing PD from Controls were the independent variables. Gender and age at baseline were used as covariates.* Observed differences in cortical thickness were then correlated to other potential disease-related measures (i.e., UPDRS-III, MoCA, CSF α -syn, and tau and A β ₄₂ markers) using Pearson partial correlations *as post-hoc analysis. All the analyses were performed using MatLab 2015b statistics toolbox.”*

Results: *“We then performed a linear regression model to examine whether cortical thinning was related to cognitive decline (as assessed by MoCA) and found the overall model, **controlling for covariates**, was significant ($r^2=0.12, p=0.03$).”*

Figure Legend: *“Figure 4. (a) Linear regression model between changes (delta) in cortical thickness clusters and delta cognitive score (MoCA). The overall model was significant; specifically, thinning of the left frontal cluster ($\beta=-0.23, p=0.04$) **and age ($\beta=0.22, p=0.01$)** were significantly related to decline in cognitive performance.”*

Reviewer #3 (Remarks to the Author):

The authors have made good effort addressing my questions. I have two more questions.

(1) Because the authors used all 15 subcortical structures as the disease reservoir, the sum of connectivity x atrophy to all the 15 regions was used to relate to cortical thinning. Such approach might not be sensitive to the specific contribution of each disease onset region. It would be important and informative to compare the predictive value of these 15 regions. Or the authors could take the maximum value among the 15 regions and perform correlation analysis with cortical thinning.

If the propagation hypothesis is correct, then different areas will contribute differently based on their content of misfolded synuclein (for which atrophy is a proxy) and connectivity. We have accounted for this in our disease exposure formula which inherently

weights different subcortical structures based on their atrophy level.

$$Disease\ Exposure(i) = \sum_j Conn_{ij} \cdot Atrophy(j)$$

Since area j has an atrophy level corresponding to the level it is affected by PD, it will contribute to the disease exposure measure more than another region k which has lower level of atrophy at baseline. For example, basal ganglia regions have 2-3 time greater atrophy than the hippocampus, and so should contribute 2-3 fold more to disease exposure. We think the model used is the correct test of the propagation hypothesis (see below).

(2) Another concern is that striatum and brainstem might be the key region vulnerable in PD but amygdala and hippocampus/thalamus might not be. How to justify using all 15 regions as disease reservoir together? It might explain the relatively weak correlations ($r = 0.1-0.2$), which is concerned by all three reviewers here. If only striatum and midbrain are used, will the correlation results remain or even stronger? It is worth finding out.

A priori, there is no reason to believe the basal ganglia would be better propagators than the medial temporal lobe. The basal ganglia are known to be especially affected in PD because the core symptoms of tremor and bradykinesia are due to dopamine deficiency there (not necessarily intrinsic pathology). In fact, in terms of Lewy pathology, the BG might actually be less affected (at least based on Braak et al. 2003, although not on our atrophy measure). Therefore, we believe the most unbiased approach was the one used here. To go back and repeat the correlation with the strongest nodes could give rise to a criticism of circularity.

Nonetheless, to address this issue we conducted two separate models to test whether areas other than the basal ganglia (e.g., amygdala, hippocampus, and thalamus) might dampen the effects and whether areas strongly implicated in PD (e.g., those within basal ganglia) might strengthen our effects. The results for functional and structural connectivity are summarized in Table Sup2 (functional connectome) and Table Sup3 (Structural connectome), respectively.

These show that the correlations from only amygdala, hippocampus, thalamus are weaker than those from basal ganglia. This may indicate greater contribution from the basal ganglia, although we do not think these findings unequivocally prove this. We add a brief section in the paper to mention this result. We think that the stronger correlations with the basal ganglia only model strengthen the spread hypothesis.

“As a follow-up exploratory analysis, we tested whether striatal and non-striatal regions within our disease reservoir may contribute differently to disease exposure. Based on functional connectivity, analysis restricted to a basal ganglia reservoir showed an increased effect size within the left hemisphere ($r=0.29$, $p<0.0001$, $r \in [0.16,0.40]$, $p_perm<0.0001$), right hemisphere ($r=0.37$, $p<0.0001$, $r \in [0.26,0.47]$, $p_perm<0.0001$) and across the whole-brain ($r=0.29$, $p<0.0001$, $r \in [0.21,0.37]$, $p_perm<0.0001$), compared to the analysis using all affected subcortical regions. When repeating this analysis for structural connectivity, the effect size observed for a basal ganglia restricted analysis within the left hemisphere ($r=0.14$, $p=0.016$, $r \in [0.01,0.27]$, $p_perm=0.017$), right hemisphere ($r=0.15$, $p=0.012$, $r \in [0.01,0.28]$, $p_perm=0.011$) and across the whole-brain ($r=0.02$, $p=0.35$, $r \in [-0.08,0.11]$, $p_perm=0.35$) were similar to the analyses considering all subcortical structures.

We added a mention of this finding in the Discussion:

“ Finally, the disease exposure analysis limited to the basal ganglia only yielded greater effects on cortical thinning than the whole subcortical reservoir model. This might suggest a greater contribution of basal ganglia to cortical propagation, although our results do not unequivocally demonstrate this.”

Table Sup2. Summary table of Spearman correlations between disease exposure based on functional connectivity and progression.

	r-value	CI 95% lower	CI 95% upper	p-value	p-perm
Disease Exposed based on Functional Connectivity to All Subcortical Structures					
bilateral	0.19	0.11	0.28	0.0000	0.0000
right hemisphere	0.27	0.16	0.37	0.0000	0.0000
left hemisphere	0.22	0.09	0.33	0.0005	0.0004
Disease Exposed based on Functional Connectivity to Amygdala, Hippocampus, and Thalamus					
bilateral	-0.07	-0.16	0.02	0.06	0.058
right hemisphere	-0.03	-0.15	0.09	0.32	0.33
left hemisphere	-0.02	-0.14	0.11	0.38	0.39
Disease Exposed based on Functional Connectivity to Basal Ganglia					
bilateral	0.29	0.21	0.37	0.0000	0.0000
right hemisphere	0.37	0.26	0.47	0.0000	0.0000
left hemisphere	0.29	0.16	0.40	0.0000	0.0000

Table Sup3. Summary table of Spearman correlations between disease exposure based on structural connectivity and progression.

	r-value	CI 95% lower	CI 95% upper	p-value	p-perm
Disease Exposed based on Structural Connectivity to All Subcortical Structures					
bilateral	-0.03	-0.13	0.06	0.23	0.75
right hemisphere	0.12	-0.02	0.25	0.04	0.04
left hemisphere	0.14	0.00	0.28	0.01	0.01
Disease Exposed based on Structural Connectivity to Amygdala, Hippocampus, and Thalamus					
bilateral	-0.12	-0.21	-0.03	0.004	0.005
right hemisphere	-0.02	-0.14	0.11	0.38	0.40
left hemisphere	0.15	0.02	0.28	0.01	0.011
Disease Exposed based on Structural Connectivity to Basal Ganglia					
bilateral	0.02	-0.08	0.11	0.35	0.35
right hemisphere	0.15	0.01	0.28	0.012	0.011
left hemisphere	0.14	0.006	0.27	0.016	0.017

References:

- 1 Caspell-Garcia, C. *et al.* Multiple modality biomarker prediction of cognitive impairment in prospectively followed de novo Parkinson disease. *PLOS ONE* **12**, e0175674, doi:10.1371/journal.pone.0175674 (2017).
- 2 Pereira, J. B. *et al.* Initial cognitive decline is associated with cortical thinning in early Parkinson disease. *Neurology* **82**, 2017-2025, doi:10.1212/wnl.0000000000000483 (2014).

REVIEWERS' COMMENTS:

Reviewer #2 (Remarks to the Author):

The authors have addressed my remaining concerns.

Reviewer #3 (Remarks to the Author):

The authors have adequately addressed my concerns. No further questions.